# Dual mode of embryonic development is highlighted by expression and function of *Nasonia* pair-rule genes

Miriam I Rosenberg[1][†]*, Ava E Brent[1][‡], François Payre[2,1], Claude Desplan[1]*

[1]Center for Developmental Genetics, Department of Biology, New York University, New York, United States; [2]Centre de Biologie du Développement, UMR5547 CNRS/Université de Toulouse, Toulouse, France

**Abstract** Embryonic anterior–posterior patterning is well understood in *Drosophila*, which uses 'long germ' embryogenesis, in which all segments are patterned before cellularization. In contrast, most insects use 'short germ' embryogenesis, wherein only head and thorax are patterned in a syncytial environment while the remainder of the embryo is generated after cellularization. We use the wasp *Nasonia* (*Nv*) to address how the transition from short to long germ embryogenesis occurred. Maternal and gap gene expression in *Nasonia* suggest long germ embryogenesis. However, the *Nasonia* pair-rule genes *even-skipped*, *odd-skipped*, *runt* and *hairy* are all expressed as early blastoderm pair-rule stripes and late-forming posterior stripes. Knockdown of *Nv eve*, *odd* or *h* causes loss of alternate segments at the anterior and complete loss of abdominal segments. We propose that *Nasonia* uses a mixed mode of segmentation wherein pair-rule genes pattern the embryo in a manner resembling *Drosophila* at the anterior and ancestral *Tribolium* at the posterior.

**\*For correspondence:** cd38@nyu.edu (CD); miriamr1@tx.technion.ac.il (MIR)

**Present address:** [†]The Ruth and Bruce Rappaport Faculty of Medicine, Technion-Israel Institute of Technology, Haifa, Israel; [‡]Department of Biological Sciences, Columbia University, New York, United States

**Competing interests:** The authors declare that no competing interests exist.

**Reviewing editor**: Duojia Pan, HHMI, Johns Hopkins University, United States

## Introduction

Control of axial patterning and embryonic development is well understood in *Drosophila* (reviewed in *Liu and Kaufman, 2005b*; *Peel et al., 2005*; *Rosenberg et al., 2009*; *Pankratz and Jaj, 1993*). Extensive work has elucidated the genetic basis of establishment of the anterior–posterior (A–P) and dorsal–ventral (D–V) axes of the fly embryo. For the A–P axis, maternally loaded mRNAs generate localized signaling centers at each pole of the egg to establish morphogenetic gradients. These gradients instruct, in a concentration dependent manner, broad domains of expression of early zygotic genes, the 'gap genes' (*Chen et al., 2012*). This is made possible in part by the syncytial environment of the early blastoderm embryo where nuclei are not bounded by membranes, allowing diffusion of morphogen transcription factors through a shared cytoplasm without the need for cell–cell signaling. In this environment, broad activation by maternal factors coupled with repressive activities by the gap genes leads to the expression of the pair-rule genes in two-segment periodicity, as pair-rule stripes. The overlapping registers of different pair-rule genes ultimately establish segment polarity through activation of the segment polarity genes, each expressed in stripes with single segmental register. This mode of development is termed 'long germ' embryogenesis because the embryo occupies all of the blastoderm apart from a dorsal region representing the extraembryonic ammnioserosa. A striking feature of long germ embryogenesis is that virtually all of segment patterning is completed synchronously in the syncytial environment. However, forays into other insect models have revealed that the *Drosophila* paradigm is an evolutionarily derived state, and that insects generally undergo a very different type of embryogenesis and segmentation (reviewed in *Liu and Kaufman, 2005b*; *Peel et al., 2005*; *Rosenberg et al., 2009*).

Unlike flies, most insect embryonic primordia occupy only a small portion of the blastoderm and only few anterior segments (head and thorax) are patterned in a syncytial environment. The remainder

**eLife digest** Networks of genes that work together are widespread in nature. The conservation of individual genes across species and the tendency of their networks to stick together is a sign that they are working efficiently. Furthermore, it is common for existing gene networks to be adapted to perform new tasks, instead of new networks being invented every time a similar but distinct demand arises. One important question is: how can evolution use the same building blocks—such as the genes in a functioning network—in different ways to achieve new outcomes?

The gene network that sets up the 'body plan' of insects during development has been well studied, most deeply in the fruit fly, *Drosophila*. Like all insects, the body of a fruit fly is divided into three main parts—the head, the thorax and the abdomen—and each of these parts is made up of several smaller segments. There is a remarkable diversity of insect body plans in nature, and yet, these seem to arise from the same gene networks in the embryo.

When a *Drosophila* embryo is growing into a larva, all the different body segments develop at the same time. In most other insects, however, segments of the abdomen emerge later and sequentially during the development process. The ancestors of most insects are also thought to have developed in this way, which is known as 'short germ embryogenesis'. So how did the so-called 'long germ embryogenesis', as observed in *Drosophila*, evolve from the short germ embryogenesis that is observed in most other insects?

The gene network that controls development includes the 'pair-rule genes' that are expressed in a pattern of alternating stripes that wrap around, top to bottom, along most of the length of the embryo. These stripes mark where the edges of each body segment will eventually develop. In fruit flies, this pattern extends along the entire length of the embryo and the stripes all appear at one time. However, in the abdominal region of short germ insects, the pair-rule genes are expressed in waves that pass through the posterior region as it grows, with new segments being added one behind the other.

Now, Rosenberg et al. have attempted to explain how the same genes can be used to direct the segmentation process in such different ways by studying another long germ insect species, the jewel wasp. Analysis of the expression of pair-rule genes in the jewel wasp shows that it uses a mixed strategy to control segmentation. The development of segments at the front of its body is directed in the same way as the fruit fly, with all these segments laid down together. However, the segments at the rear of the body are only patterned later, one after the other, like most other insects.

The work of Rosenberg et al. suggests that the jewel wasp represents an intermediate step between ancestral insects and *Drosophila* in the evolution of the gene network that patterns the 'body plan'. Identifying and studying these intermediate forms allows us to understand the ways in which evolution can innovate by building upon what has come before.

of the embryo is generated after cellularization via a 'growth zone', at the posterior region of the embryo. This mode is termed 'short germ' embryogenesis. Recently, the mechanisms governing posterior segment patterning and growth in the *Tribolium* embryo were characterized in elegant detail (**Choe et al., 2006**; **Choe and Brown, 2009**; **El-Sherif et al., 2012**; **Sarrazin et al., 2012**): Oscillations of the pair-rule gene *Tc'odd-skipped* (*Tc'odd*) in the growth zone are in turn linked to a circuit of two other pair-rule genes, *Tc'runt* and *Tc'even-skipped* (*Tc'eve*), such that each new pair of segments experiences a pulse of *Tc'odd* and requires both *Tc'eve* and *Tc'runt* expression in order to progress; the driver of these oscillations is still unknown. The waves of expression of *Tc'odd-skipped* pass through the growth zone rhythmically, generating segments and new stripes of stable expression with each periodic pulse (**Sarrazin et al., 2012**). RNAi of *Tc'odd*, *Tc'runt*, or *Tc'eve* results in asegmental embryos, underscoring their requirement in both growth zone-derived segments and earlier blastoderm anterior segments (**Choe et al., 2006**). In contrast, the pair-rule genes *Tc'sloppy paired* (*Tc'slp*) and *Tc'paired* (*Tc'prd*) appear in two-segment periodicity in head stripes and in stripes that emerge from the growth zone, and RNAi of those genes produce classical pair-rule phenotypes, in which alternating segments are lost (**Choe and Brown, 2007**). Live imaging revealed that formation of posterior segments results primarily from convergent extension and short-range cell movements and not strictly from cell division within the 'growth zone'. This mechanism appears similar to both segmentation of vertebrate presomitic mesoderm (reviewed in [**Dubrulle and Pourquie, 2004**] and [**Pourquie, 2011**]) and to segment formation

in more basal arthropods, including the centipede *Strigamia maritima* (**Chipman et al., 2004**; **Chipman and Akam, 2008**) and the spider *Cupiennius salei* (**Stollewerk et al., 2003**), suggesting it as an ancient mechanism inherited from the last common ancestor of all segmented animals (though this interpretation is still debated; reviewed in [**Davis and Patel, 1999**]).

As *Drosophila* is only one example of a derived long germ strategy, one outstanding question is how transitions from short germ to long germ embryogenesis occurred, such that the same set of segmentation genes possesses different functions. The careful study of additional long germ insects should shed light on what aspects of *Drosophila* development are essential facets of long germ embryogenesis and which aspects are more evolutionarily labile. Other model species have been studied, including long germ beetles (e.g., *Callosobruchus* order: *Coleoptera*; (**Patel et al., 1994**)), and several members of the order *Hymenoptera*, including the honeybee, *Apis mellifera* (**Dearden et al., 2006**; **Wilson et al., 2010**; **Wilson and Dearden, 2011**, **2012**) and the jewel wasp, *Nasonia vitripennis* (*Nv*) (**Pultz et al., 1999**; **Werren et al., 2010**). However, systematic characterization of their pair-rule genes and segmentation mechanisms is still incomplete.

We use the wasp *Nasonia vitripennis* as a model for the study of A–P patterning, as a species that appears to have evolved, independently of *Drosophila*, a similar mode of long germ embryogenesis. We have previously characterized the early patterns of *Nasonia* segmentation genes and found that maternal and gap gene expression confirms a long germ mode of embryogenesis. This conclusion was based on the existence of two polar signaling centers, each utilizing localized maternal *Nv orthodenticle (otd)* mRNA that encodes a morphogen. *Nv otd* acts in combination with *Nv hunchback* (*hb*) and localized maternal *Nv giant (gt)* at the anterior, and with localized maternal *Nv caudal (cad)* at the posterior, to specify positional identity. The domains of zygotic expression of *Nv hb*, *Nv gt*, *Nv cad*, *Nv Krüppel* (*Kr*), *Nv tailless* (*tll*), and *Nv knirps* (*kni*) closely resemble their *Drosophila* counterparts, consistent with a similar mode of blastoderm allocation (**Pultz et al., 2005**; **Lynch et al., 2006**; **Olesnicky et al., 2006**; **Brent et al., 2007**). Although these data support *Drosophila*-like early regulatory interactions and a long germ mode of embryogenesis, little was known about later stages of *Nasonia* embryonic patterning.

We analyzed the expression and function of the pair-rule genes *Nv eve*, *Nv odd*, *Nv runt* and *Nv h* during embryogenesis. We found that each gene is expressed in both a canonical long-germ pair-rule stripe pattern at the anterior, as well as late-forming posterior stripes, indicating a dual mode of regulation. Strikingly, *Nv eve* is ultimately expressed in a total of 16 segmental stripes, of which six are derived from a single posterior stripe in the cellularized blastoderm. We also observe waves of *Nv odd* expression that resemble the waves of *Tribolium odd* expression, suggesting the residual activity of a segmentation clock in *Nasonia*. As in *Tribolium*, we found that mitoses do not occur exclusively at the site of late forming segments, but mitotic figures are not randomly distributed throughout the embryo. Instead, coordinated mitoses resembling the later mitotic domains of *Drosophila* (**Foe, 1989**) appear and progress in waves from anterior to posterior, and are largely excluded from stripes of *eve* expression, suggesting a coordination of mitoses by segmentation genes. Using morpholinos to knock down gene function, we found that *Nv eve, Nv odd* and *Nv h* phenotypes do not affect alternating segments at the posterior, unlike what is observed in *Drosophila*. Instead, these 'pair-rule' genes are required for the formation of a continuous posterior region comprising abdominal segments A5–A10. Phenotypes in the anterior of the embryo are gene-specific; each gene exhibits a partial pair-rule phenotype in the allelic series. We suggest that *Nasonia* uses 'pair-rule' genes to pattern the embryo in a manner that resembles both *Drosophila* and *Tribolium*. We present a model for how this mixed mode of segmentation is achieved.

## Results

### *Nasonia* even-skipped exhibits character of both long and short germ patterning

The expression of eve has been studied in many insects, owing to a widely cross-reacting antibody. Its promoter has also been well studied in *Drosophila* and has become a classic example of modular gene control (**Patel et al., 1992**; **Small et al., 1992**, **1996**). We used circular RACE from total embryo RNA (**McGrath, 2011**) to generate a fragment of approximately 1 kb corresponding to the coding region of *Nv eve*, including the highly conserved homeodomain (Genbank Accession# KC168090). Several minor transcript variants were captured and sequenced, but not studied further (see 'Materials and methods' for GenBank accession numbers). The homeodomain of *Nv* Eve shares 81.7% amino acid identity with its *Drosophila* counterpart.

We used in situ hybridization to look at the expression pattern of *Nv eve* during *Nasonia* embryogenesis (*Figure 1*). *Nv eve* expression becomes detectable as a broad early domain in the blastoderm embryo at around 3 hr after egg laying (AEL) (*Figure 1A*). This domain broadens and its boundaries sharpen between 3 and 4 hr, by which time a faint posterior stripe (hereafter referred to as 'stripe 6', see below) becomes evident (*Figure 1B,C*). As embryogenesis progresses toward cellularization, the anterior domain splits into three distinct double-segment periodicity pair-rule stripes, stripes 1, 2 and 3 (*Figure 1D–F*). By cellularization, around 6 hr, a faint 4/5 stripe appears between the anterior stripes and stripe 6, which has become more intense (*Figure 1F*). Between 6 and 8 hr AEL, stripe 4/5 splits into distinct double-segment periodicity stripes 4 and 5, whereas stripes 1 and 2 split into single-segment periodicity segmental stripes (*Figure 1G–I*). In *Drosophila*, eve secondary stripes form de novo, between primary pair-rule stripes, in contrast to secondary *paired* stripes that later split from primary stripes, forming segmental stripes that affect all segments (*Macdonald et al., 1986*; *Kilchherr et al., 1986*). Splitting of *Nasonia eve* double-segment stripes into single-segment stripes may occur by a similar mechanism (see below). As gastrulation progresses between 8 and 10 hr AEL, double-segment pair-rule stripes 3–5 also split to give rise to two distinct single-segment stripes each (*Figure 1J–L*). This anterior to posterior progression of *Nv eve* stripes is consistent with the sequential appearance of the segment polarity genes *Nv wg* (*Figure 1—figure supplement 1*) and *Nv en* (*Pultz et al., 1999*), which are first detected around cellularization in a few anterior segments and then appear in stripes progressively, in an anterior to posterior manner.

A remarkable feature of *Nv eve* expression is that the posterior stripe 6 broadens significantly at about 10 hr AEL (*Figure 1L–N*), before generating six additional stripes with single-segment periodicity, allowing the embryo to reach 16 individual segmental stripes by the time the germ band is fully extended. This division of stripe 6 initiates with an anterior band that will give rise to four stripes (segmental stripes 11–14; *Figure 1P,Q*, arrowheads; see below), and two later-appearing segmental stripes 15 and 16 (*Figure 1Q*, arrowheads). The last stripe, *Nv eve* 16, appears only at full germ band extension (*Figure 1R*) completing the 16 stripes observed at this stage.

The *Nasonia* embryo has 16 segments, whereas *Drosophila* has only 14 (*Figure 2A*). In *Drosophila*, *eve* and other pair-rule genes are expressed with double-segment periodicity: seven transverse 'pair-rule' stripes are evident as a full complement in the blastoderm embryo at cellularization. If the *Nasonia* embryo were patterned using the same mechanisms as *Drosophila*, then eight pair-rule stripes would be predicted. However, only five truly pair-rule (double segment) stripes are apparent at cellularization, while stripe 6 gives rise later to four, then six single-segment stripes and six segments and is therefore not pair-rule (*Figure 2E*). This delayed sequential posterior segmentation is therefore more reminiscent of the segmentation described in short germ insects.

## Control of Nv eve by gap genes

To explore this apparent combination of short and long germ characters, we determined how *Nv eve* expression is controlled by upstream genes in the known *Nasonia* A–P patterning network. Early embryonic expression of *Drosophila eve* is controlled by maternal and gap genes, including *bicoid (bcd)*, *hb*, *Kr*, *gt*, *kni* and *torso* (*MJaJ & H, 1993*; *Small et al., 1992*, *1996*; *Schroeder et al., 2004*; *Small and Levine, 1991*), whereas later maintenance is achieved via autoregulation (*Jiang et al., 1991*). Some *Tribolium eve* pair-rule stripes are also under the control of gap genes although some of the segments themselves are born much later than gap gene expression (*Sulston and Anderson, 1996*; *Cerny et al., 2005*). For example, in *Tc'Kr* mutant embryos, segments anterior to the normal *Kr* expression domain (T1–T3) appear wild type, but expression of both *Tc'eve* and *Tc'en* is lost in posterior segments and no segments are formed posterior to A4 (*Cerny et al., 2005*). Therefore, *Tribolium* gap genes can affect the specification of segments that are not yet formed, presumably because of interactions with the growth zone. In other short germ insects, like *Oncopeltus fasciatus, eve* acts as a gap gene, regulating expression of *hunchback* and *Krüppel* (*Liu and Kaufman, 2005a*).

To determine how the known maternal and gap genes regulate *Nv eve* expression in the early embryo, we used parental RNAi injections in pupal *Nasonia* females to knockdown *Nv gt, Nv Kr, Nv tll* and *Nv cad* mRNA, as well as a null mutation in *Nv hb* (*Pultz et al., 2000*, *2005*).

## Nv gt

As we previously reported, *Nv giant* knockdown results in the loss of all segments anterior to A1 and fusion of segments A6 and A7 (*Brent et al., 2007*). *Nv gt* RNAi blastoderm embryos exhibit the loss

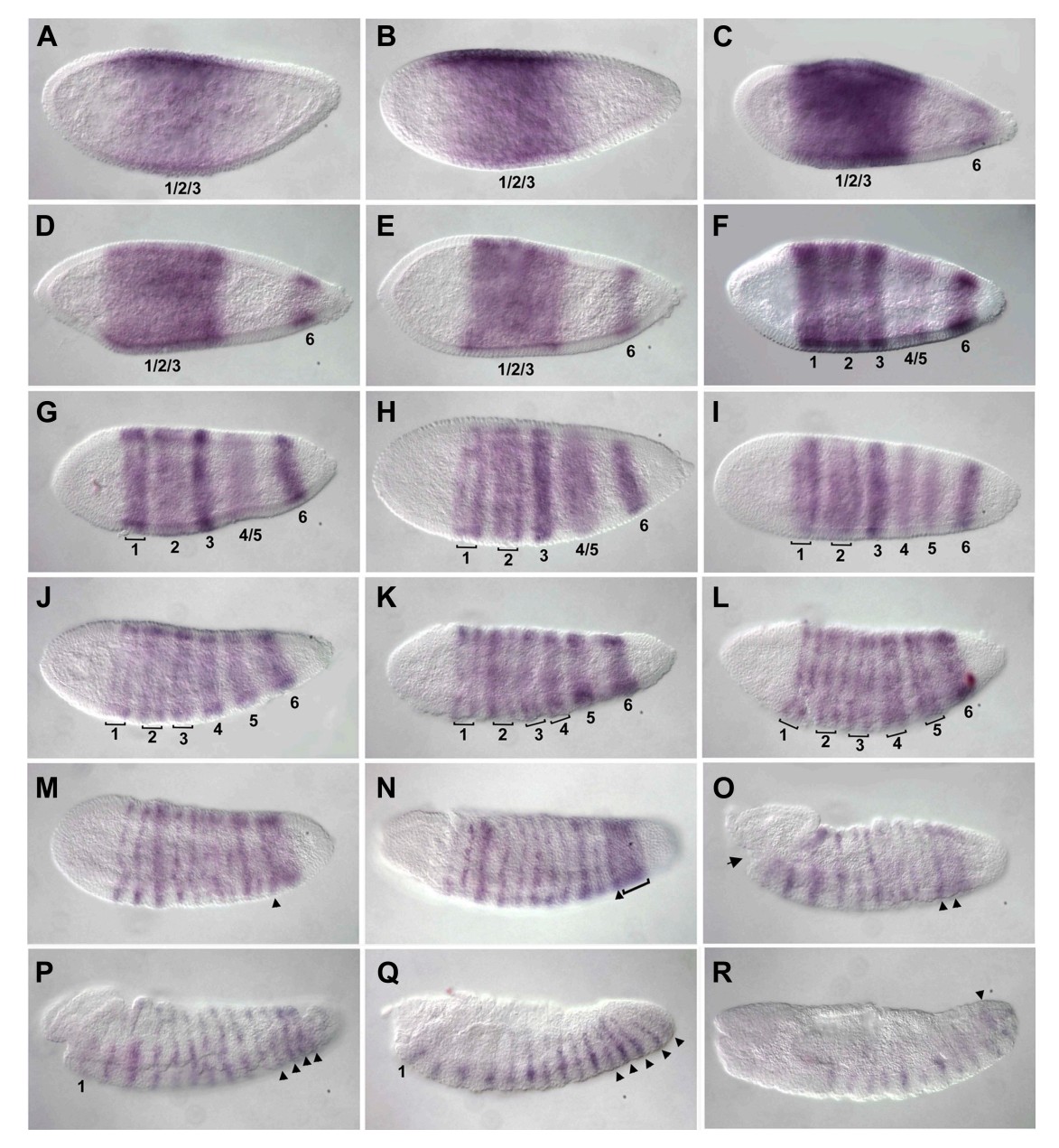

**Figure 1**. Summary of *Nasonia eve* mRNA expression. Embryos are shown with anterior left and dorsal up. *Nv eve* is initially expressed in a broad domain (**A** and **B**), which sharpens as a posterior stripe becomes visible at around 4 hr after embryo laying (AEL) (**C** and **D**). The broad domain retracts anteriorly and gives rise to three apparently double-segment stripes (**E** and **F**). Between stripes 3 and posterior stripe 6, an additional double stripe precursor comes up at around 6 hr AEL (stripe 4/5; panels **F** and **G**) and this splits to form two double-segment stripes, '4' and '5' as double-segment stripes 1–3 split into two single-segment stripes each between 6 and 8 hr AEL (**F**–**J**). Stripes 4 and 5 also split to form single-segment stripes during early gastrulation, and stripe 6 broadens (**K** and **L**), giving rise to stripes that are visibly distinct during germ band extension in non-fluorescent staining by 10–12 hr AEL (**M**–**R**, arrowheads). There are a total of 16 single-segment stripes of *Nv eve*.

The following figure supplements are available for figure 1:

**Figure supplement 1**. *Nv wingless (Nv wg)* mRNA expression in the embryo.

of *Nv eve* double-segment stripes 1–3 (***Figure 2F***), as well as aberrant resolution of the first splitting events of stripe 6 (***Figure 2G***, arrowheads). Double in situ hybridization shows that, in the wild type, a late posterior stripe of *Nv gt* forms after *Nv eve* stripe 6 and appears to be within the *Nv eve* stripe

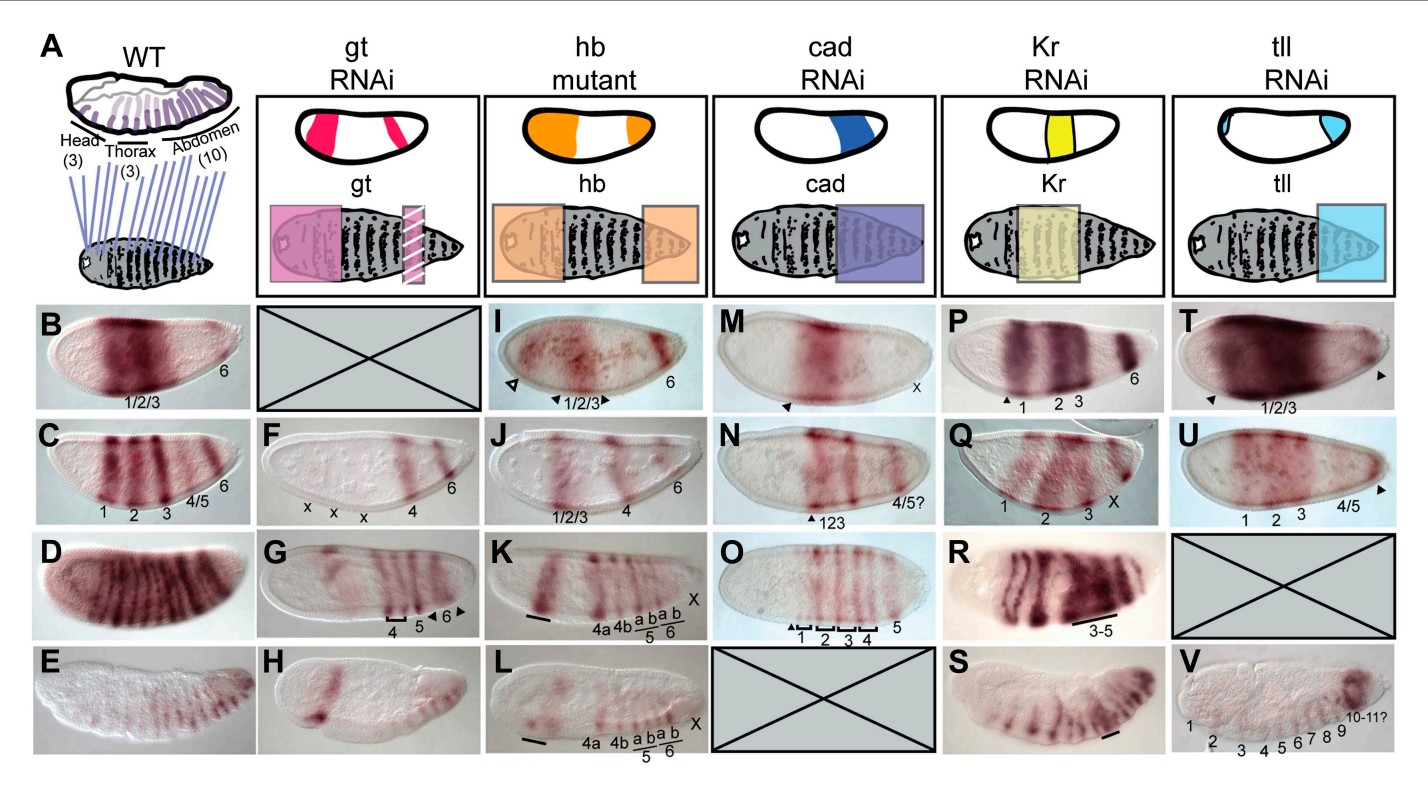

**Figure 2**. *Nv eve* epistasis with maternal and gap genes. (**A**) Schematic representation of the germ-band-extended embryo, showing 16 single-segment stripes of *Nv eve* expression, and their segment counterparts in the patterned larval cuticle. Colored boxes cover the segments of the larval cuticle that are lost or fused in each RNAi background. All embryos are shown anterior left, dorsal up (except where indicated). *Nv eve* mRNA expression is shown in each embryo (**B–G**). Wild-type (WT) embryos are shown as staged controls for RNAi embryos. (**B**) WT early blastoderm embryo. (**C**) WT cellular blastoderm embryo. (**D**) WT early gastrula extension embryo. (**E**) WT germ-band-retracted embryo. (**F–H**) *gt* RNAi embryos stained for *Nv eve* mRNA expression. (**F**) Cellular blastoderm embryo with reduced *Nv gt* exhibits loss of anterior *Nv eve* stripes (x). (**G**) *Nv gt* RNAi embryo in early germ-band-extension exhibits loss of anterior *Nv eve* stripes and improper splitting of *Nv eve* stripe 5, as well as aberrant dorsal anterior expression of *Nv eve*. (**H**) *Nv gt* RNAi embryo at dorsal closure exhibits a stripe of *Nv eve* at the anterior, as well as a reduced number of posterior segmental *Nv eve* stripes. (**I–L**) *Nv hb* mutant embryos stained for *Nv eve* mRNA expression. (**I**) Early blastoderm *Nv hb* mutant embryos have a reduced central *Nv eve* domain (bounded by black arrowheads), and an ectopic anterior *Nv eve* stripe (white arrowhead). (**J**) *Nv hb* mutant cellular blastoderm embryo with a single anterior domain of *Nv eve* that has failed to resolve, and a single stripe 4 which exhibits delayed splitting. (**K**) *Nv Hb* mutant germ-band extension embryo with fused anterior domain (line) and 6 segmental stripes, representing derivatives of *Nv eve* stripes 4 and 5 and two derivatives of stripe 6; additional stripe 6 derivatives are absent (x). (**L**) *hb* mutant dorsal closure embryo exhibiting fused anterior domain (line) and the same number of derivatives as in (**M**), with more posterior segments missing (x). (**M–O**) *Nv cad* RNAi embryos stained for *Nv eve* mRNA expression. (**M**) *Nv cad* RNAi early blastoderm with reduced central *Nv eve* domain that is also posteriorly shifted (anterior boundary indicated by black arrowhead). (**N**) *Nv cad* RNAi cellular blastoderm embryo with posteriorly shifted (arrowhead), reduced *Nv eve* central domain, whose splitting is delayed. (**O**) *Nv cad* RNAi early gastrula embryo with posterior shift in *Nv eve* expression (black arrowhead). Four double-segment periodicity stripes are split into single-segment stripes and stripe 5 remains intact. (**P–S**) *Nv Kr* RNAi embryos stained for *Nv eve* mRNA expression. (**P**) *Nv Kr* RNAi precellular blastoderm embryo with aberrant *Nv eve* central domain resolution, where stripes 2–3 appear posteriorly shifted. (**Q**) Dorsolateral view of a *Nv Kr* RNAi embryo where stripes 2 and 3 are less refined than WT and 3 is posteriorly shifted. No stripe 4/5 expression is detected (X). (**R**) *Nv Kr* RNAi early gastrula embryo with aberrant stripe 2 splitting and aberrant resolution of stripes 3–5. (**S**) Moderately affected *Nv Kr* RNAi germ-band retraction embryo with fused segments in the middle of the embryo (line). (**T–V**) *Nv tll* RNAi embryos stained for *Nv eve* mRNA expression. (**T**) *Nv tll* RNAi early blastoderm embryo with expanded *Nv eve* expression domains toward both poles (arrowheads). (**U**) *Nv tll* RNAi precellular blastoderm embryo showing delayed resolution of *Nv eve* stripes 1–3 and *Nv eve* stripe 6 shifted to the extreme posterior pole of the embryo (arrowhead). (**V**) *Nv tll* RNAi dorsal closure embryo showing abnormal posterior *Nv eve* stripe formation.

The following figure supplements are available for figure 2:

**Figure supplement 1**. *Nv eve/Nv gt* double FISH in the embryo.

**Figure supplement 2**.

6 domain (*Figure 2—figure supplement 1*). These data suggest that a posterior *Nv gt* domain may partially affect stripe 6 splitting. Late *Nv gt* RNAi embryos exhibit strong anterior defects after dorsal closure, but the other posterior single-segment stripes of *Nv eve* appear unaffected (*Figure 2H*).

## Nv hb

In *Nv hb* mutant (*headless*) embryos, both head and thoracic fates and abdominal fates posterior to A6 are lost (*Pultz et al., 2005*). Consistent with this phenotype, *Nv eve* double-segment stripes 1–3 (that give rise to head and thoracic fates) form but never resolve (*Figure 2K*, black arrowheads). An ectopic stripe of *eve* can be seen in the anterior of some embryos (*Figure 2I*, white arrowhead), consistent with the ectopic stripe of *Nv cad* (an activator of *eve*) in the head of some *Nv hb* mutant embryos (*Olesnicky et al., 2006*). Gastrula and later germ band embryos exhibit normal *Nv eve* double-segment stripe formation and splitting of stripes 4 and 5, which give rise to segments A1–A4. However, only the two anteriormost segments are formed from *Nv eve* stripe 6 (6a and 6b) in *Nv hb* mutant embryos (*Figure 2K,L*).

## Nv caudal

*Nv caudal* (*cad*) is expressed maternally as a mRNA gradient with a localized posterior source (*Olesnicky et al., 2006*). *Nv cad* RNAi results in loss of all segments posterior to A1. Moderately affected *Nv cad* RNAi embryos exhibit a reduced early broad domain of *Nv eve* that is slightly shifted posteriorly (*Figure 2M*). This domain resolves poorly, with only weak activation of anterior *Nv eve* stripes and no posterior abdominal expression of *Nv eve* (*Figure 2N–O*).

## Nv Kr

As in *Drosophila*, *Nv Krüppel* (Kr) is expressed in a central domain, and *Nv Kr* is required for formation of segments T3 to A4 (*Brent et al., 2007*). In *Nv Kr* RNAi embryos, both anterior and posterior domains of *Nv hb* expression expand towards the center of the embryo (*Brent et al., 2007*). Consistent with expansion of *Nv hb*, we observed that *Nv eve* stripe 2 and 3 exhibit aberrant resolution and *Nv eve* stripes 4 and 5 fail to resolve in embryos with knocked-down *Nv Kr* (*Figure 2P–S*). Posterior segments are unaffected, as reflected by normal expression of *Nv eve* posterior to stripe 5 (segment A4; *Figure 2R,S*). This phenotype is dramatically different from *Tc'Kr* knockdown where all posterior segments are deleted, likely because *Nv Kr* is expressed anterior to the growth zone while *Tc'Kr* abuts it.

## Nv tll

*tailless* mRNA is expressed in both an anterior and a posterior domain, though only posterior segments are affected by *Nv tll* RNAi (*Lynch et al., 2006*). The most severely affected embryos are missing the six posterior abdominal segments. These embryos also exhibit an apparent slight anterior shift of the broad early domain of *Nv eve* expression and of stripe 6 (*Figure 2T,U*). Stripe 6 does not appear to resolve, resulting in an enduring ring of *Nv eve* expression and no *Nv eve* single-segment stripes posterior to this ring are apparent (*Figure 2V*).

Taken together, and consistent with previously described cuticular phenotypes for maternal and gap genes in *Nasonia* (*Pultz et al., 2005*; *Lynch et al., 2006*; *Olesnicky et al., 2006*; *Brent et al., 2007*; *Figure 2—figure supplement 2*), these data show that early *Nv eve* expression in blastoderm embryos involves regulatory interactions reminiscent of those underlying *Drosophila* long germ embryogenesis. However, since severe RNAi phenotypes of several genes, such as *Nv cad* and *Nv tll* results in total loss of posterior segments, these did not provide additional information for understanding the establishment of posterior *Nv eve* expression.

## *Nasonia* embryos have mitotic domains but Nv eve posterior stripe resolution does not require localized cell division

To determine whether cell division plays a role in subdivision of the *Nv eve* posterior domain into single-segment stripes, we used in situ hybridization to visualize *Nv eve* mRNA in embryos where mitotic cells were labeled with antibodies against phosphorylated histone H3. We found that there is no cell division that is consistent with a role in pair-rule stripe splitting (*Figure 3A–A"*), or in stripe 6 resolution (*Figure 3B–B"*), suggesting that the dynamics of the *Nv eve* mRNA pattern mostly involves transcriptional regulation. Nevertheless, later mitoses occur in restricted spatial domains, reminiscent of later *Drosophila* mitotic domains within segments of the expanding germ band (*Figure 3B–D"*). A relationship between gap gene function and regulation of mitotic domains via regulation of *string* has been

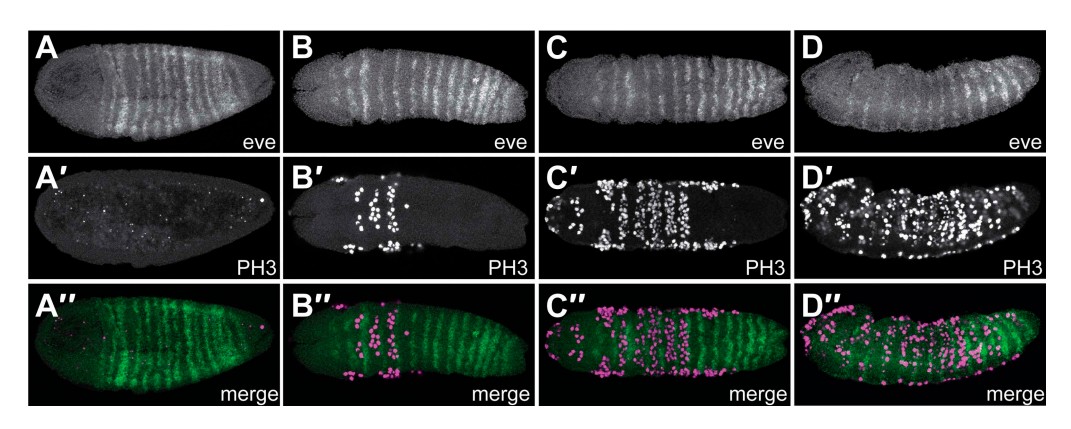

**Figure 3**. *Nv eve* expression and cell division appear to be coordinated. Embryos co-stained for *Nv eve* mRNA using in situ hybridization and fluorescent detection, as well as for mitotic figures, using an antibody against phospho Histone H3. Embryos are shown with anterior left and dorsal up, except columns **B** and **C**, which are ventral views. (**A–A"**) An early gastrula embryo exhibiting 15 stripes of *Nv eve*, including five derivatives of stripe 6 (**A**), has no evident mitotic figures in the posterior domain of *Nv eve* stripe 6 differentiation (**A'**). (**A"**) Merge of panels **A** and **A'**. (**B–D"**) Timecourse series of wild-type embryos stained for *Nv eve* mRNA and phospho-Histone H3. (**B–D**). Top panels are *Nv eve* in situ alone, middle panels (**B'–D'**) are phospho-Histone H3 antibody staining, and bottom panels (**B"–D"**) are merge images of upper panels, showing localization of mitotic figures relative to *Nv eve* stripes.

The following figure supplements are available for figure 3:

**Figure supplement 1**. Quantification of PH3 positive cells relative to *Nv eve* stripes in the embryo.

suggested in *Drosophila* (*Edgar et al., 1994*) but not demonstrated. *Nasonia* mitotic domains appear in an anterior to posterior progression, allowing for progressive expansion of segments along the A–P axis via concerted cell divisions within domains. Strikingly, mitotic figures appear to be largely excluded from *Nv eve* stripes in these early stages of embryogenesis (*Figure 3B",C"*, *Figure 3—figure supplement 1*). Later embryos in which anterior *Nv eve* stripes are beginning to fade exhibit overlap of mitotic figures with weakened *Nv eve* stripe expression (*Figure 3D"*). Ultimately, embryos exhibit widespread mitotic figures that do not correspond to any apparent concerted domains or pattern, more like the pattern of mitoses described in several short germ insects (*Handel et al., 2005*; *Liu and Kaufman, 2009*).

## Knockdown of *Nv eve* results in gap and segment polarity defects and posterior truncation

The expression of *Nv eve* suggests a combination of long germ and short germ character. To further explore this possibility, we knocked down *Nv eve* gene function. Although parental RNAi in *Nasonia* is effective for maternal and early zygotic genes (*Lynch and Desplan, 2006*; *Lynch et al., 2006*; *Olesnicky et al., 2006*; *Brent et al., 2007*), it often does not provide significant knockdown of later-acting genes. To overcome this limitation, we designed an *Nv eve* morpholino overlapping the translation start site, as well as one directed at the exon–intron junction in the homeobox. These two independent morpholinos are expected to disrupt *Nv eve* activity and indeed result in comparable phenotypes.

The *Nasonia* larval cuticle has relatively few landmarks to allow for interpretation of segmentation phenotypes. Beyond the denticle belts present on each of the three thoracic segments and ten abdominal segments, large spiracles are found on segments T2, A1, A2 and A3 (*Figure 4A*, yellow arrowheads). In the head, two bright structures indicate the positions of antennal papillae. Morpholino block of *Nv eve* causes a range of phenotypes (*Figure 4B–E*), resulting in severe truncation of the embryo with loss of posterior-derived segments as well as a partial pair-rule phenotype for more anterior segments. The phenotypic series includes progressive truncation at the posterior, causing fusion of segments A9–10 in the least affected cuticles, and then A8–10 (*Figure 4C*) with segment A6 eventually

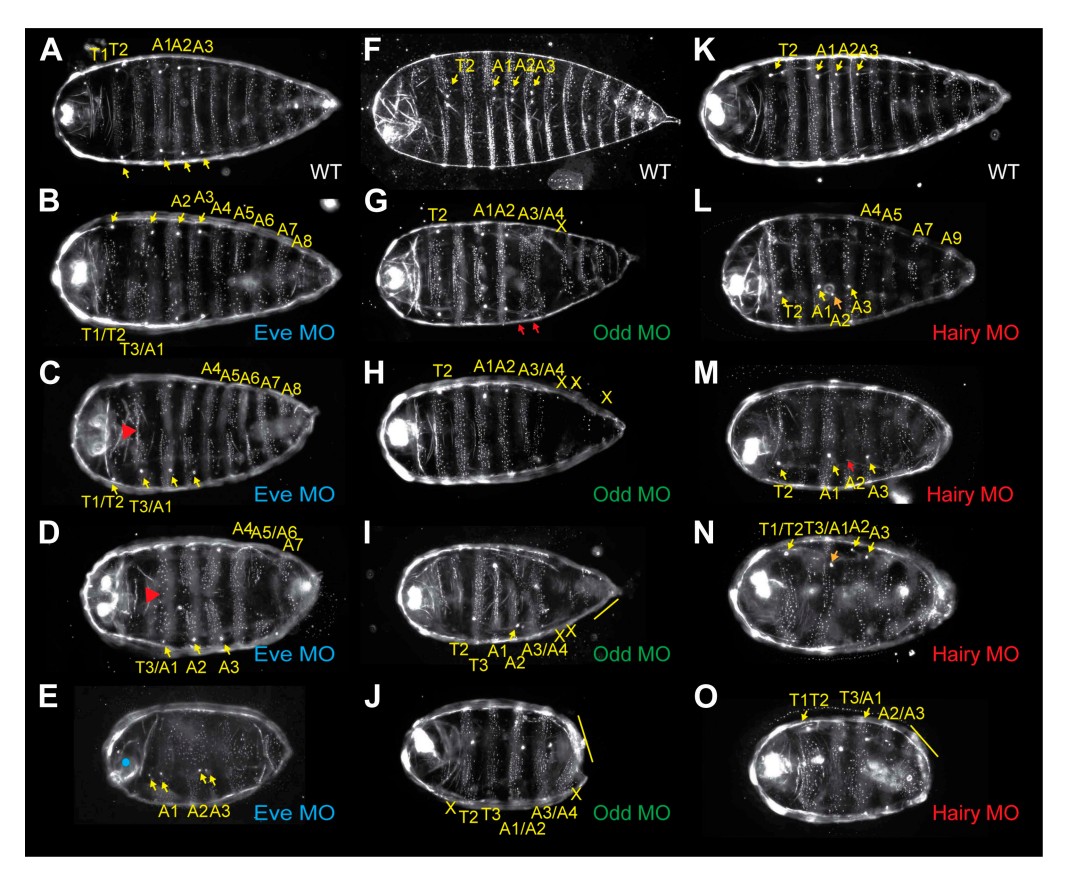

**Figure 4**. Morpholino knockdown of *Nv eve*, *Nv hairy*, and *Nv odd* results in embryo patterning defects. First instar larval cuticles are shown with anterior left and generally ventral denticle patterns are shown. (**A**, **F**, **K**) Wild-type larval cuticles. Yellow arrows indicate spiracles present on segments T2, A1, A2 and A3. Bright anterior labral appendages are apparent at the extreme anterior of the larva. (**B–E**) Unhatched larvae from *Nv eve* morpholino (MO)-injected embryos, in order of increasing phenotype severity. Red arrowheads indicate loss of midline cuticle. Blue dot indicates head open defect. Yellow arrowheads indicate position of spiracles. (**G–J**) Unhatched larvae from *Nv odd* morpholino (MO)-injected embryos, in order of increasing phenotype severity. Yellow arrows indicate position of spiracles, red arrows indicate A3/A4 fusion. X indicates naked cuticle from segment loss. Yellow line indicates multi-segment fusion. (**L–O**) Unhatched larvae from *Nv hairy* morpholino (MO)-injected embryos, in order of increasing phenotype severity. Yellow arrowheads indicate position of spiracles. Red or orange arrowheads indicate aberrantly positioned or missing spiracles. Yellow line indicates segment fusion.

lost, whereas A7 remains virtually intact. In the most severely affected embryos, the entire posterior of the embryo is truncated with A5–A10 missing. (The approximate percentage of embryos in each phenotypic class shown in *Figure 4* is indicated in the 'Materials and methods'.)

At the anterior, T1 is lost with fusion of T3 and A1 (*Figure 4B*). Segments A2 and A3 are also fused, and there is a continuous lawn of denticles from A4 to the truncated posterior. This is accompanied by a disruption of the remaining denticle belts, leaving naked cuticle along the midline (*Figure 4*, red arrows). In the most severely affected embryos, segments anterior to A1 are lost and are accompanied by head closure defects. This phenotype represents a partial pair-rule phenotype, accompanied by posterior truncation of the embryo. It also does not exhibit the lawn of denticles phenotype of strong *eve* alleles in flies (e.g., *eve*[R13] [*Macdonald et al., 1986*; *Fujioka et al., 1999*]), although severely affected *Nasonia* embryos also exhibit cuticle defects beyond a pair-rule phenotype. Hence, these results support a mixed mode of embryogenesis in *Nasonia*, with characteristic features resembling both long germ and short germ insects. To further examine this possibility, we then investigated the

expression patterns in *Nasonia* of other genes acting as pair-rule in *Drosophila,* and undertook functional characterization of their activity during embryonic development.

## Nasonia odd-skipped expression and function

In the long germ *Drosophila* embryo, *odd* is expressed with a double-segment periodicity complementary to that of *eve,* and its inactivation causes the absence of odd segments (*Nusslein-Volhard and Wieschaus, 1980*; *Coulter et al., 1990*). Its critical function as a mediator of the segmentation clock in the short germ beetle was recently elegantly described (*Sarrazin et al., 2012*). *Tc'odd* begins with blastoderm expression in double-segment periodicity stripes alternating with *Tc'eve* expression. Then, new double-segment stripes emanate from the growth zone to generate the entire complement of *odd* stripes. Secondary single-segment stripes arise later (*Sarrazin et al., 2012*).

There are three *odd* paralogs in *Nasonia,* as in flies, where they are named *odd, bowl,* and *sister of bowl* (or *sob;* (*Hart et al., 1996*)). We used sequence alignment (*Figure 5—figure supplement 2*) and phylogenetic analysis (*Figure 5—figure supplement 1*) to identify the *Nasonia* paralog that is closest to *Drosophila odd-skipped,* and refer to it hereafter as *Nv odd.* An *Nv odd* cDNA fragment comprising the region encoding the conserved DNA binding domain was used as a probe for in situ hybridization (GenBank Accession # KC142194). As observed above for *Nv eve,* the embryonic expression of *Nv odd* begins as a broad early domain in syncytial blasoderm embryos (*Figure 5A*). As this broad domain strengthens and sharpens, a ventral head patch and a posterior cap appear (*Figure 5B,C*). The broad domain resolves into two clear apparent double-segment stripes (Stripes 1 and 2, *Figure 5D*). A third double-segment stripe arises from the second stripe, expanding posteriorly (*Figure 5D–F*). At the same time, a stronger posterior domain apparently advances anteriorly. Pair-rule stripe 4 (double-segment periodicity) arises at the anterior of the first advancing 'wave' at cellularization (*Figure 5G–H*) before the posterior domain recedes again (*Figure 5I–J*). The fifth double-segment stripe arises during a second 'wave' (*Figure 5K–M*) at the onset of gastrulation. A sixth stripe arises in an apparently similar manner, though it is much fainter and appears while more posterior stripes are already differentiated (*Figure 5N,* arrowhead); the posterior cap generates two thin pair-rule stripes (*Figure 5O*) during early germ band extension. At full germ band extension, a total of eight stripes are visible (*Figure 5P*) before these fade from anterior to posterior. This dynamic expression of *Nv odd* in the posterior of the embryo is reminiscent of the waves of growth zone expression of *Tribolium odd,* where blastoderm-derived stripes initially have double-segment periodicity and later single-segment periodicity (*Choe et al., 2006*; *Sarrazin et al., 2012*).

Using double fluorescent in situ hybridization, we confirmed that the pair-rule stripes of *Nv odd* and *Nv eve* are indeed complementary to each other, although their mode of appearance is totally different. *Nv odd* double-segment stripes are posterior to, and abut each posterior single-segment stripe (i.e., 1b, 2b) from each *eve* pair-rule doublet (*Figure 6A–C*), that is, the even-numbered segmental stripes. Late forming *Nv eve* stripe 15/16 intercalates between the two *Nv odd* stripes 7 and 8 that derive from the cap, with *Nv odd* stripe 8 remaining posterior to all *Nv eve* stripes (excepting the last stripe, *eve 16,* which is the last to appear), a relationship that may have ancestral origins (see 'Discussion').

To examine the function of *Nv odd* in the embryo, we used one translation blocking and one splice blocking morpholino to knock down its expression in embryos. Inactivating *Nv odd* function leads to loss of the most posterior germ band-derived segments A5–A10 with additional anterior defects. The most sensitive phenotypes are the fusion of segments A3 and A4, and loss of segment A5 (*Figure 4G,* red arrowheads and x). More severely affected embryos exhibit loss of most segments posterior to A3/A4 and naked cuticle anterior to T2, though larval head structures are still present (*Figure 4I–J*). In many severely affected embryos, *Nv odd* knockdown causes additional loss/fusion of A1, and of T2. Thus, phenotypes comprise loss of segments T2, A1, A3 and A5 and resemble a pair-rule phenotype. A small percentage of embryos are nearly asegmental, with only small patches of denticle bands of unknown identity (not shown). Therefore, as also observed for *Nv eve* knockdown, the segmentation defects resulting from *Nv odd* inactivation only corresponds to a partial *Drosophila*-like pair-rule phenotype, and rather, in the most severe cases, resemble the phenotype of *Tc'odd* loss of function. It is of note that *Nv odd* stripes 4, 5 and 6 appear to emerge from waves of expression at the posterior of the embryo that likely specify segments A3–A5, which are most sensitive to loss of *Nv odd* function (*Figure 5E–L,* *Figure 4G*).

In summary, *Nv odd* is expressed initially in three sequentially forming anterior double-segment periodicity stripes, which appear to have *Drosophila*-like pair-rule character. Three more posterior

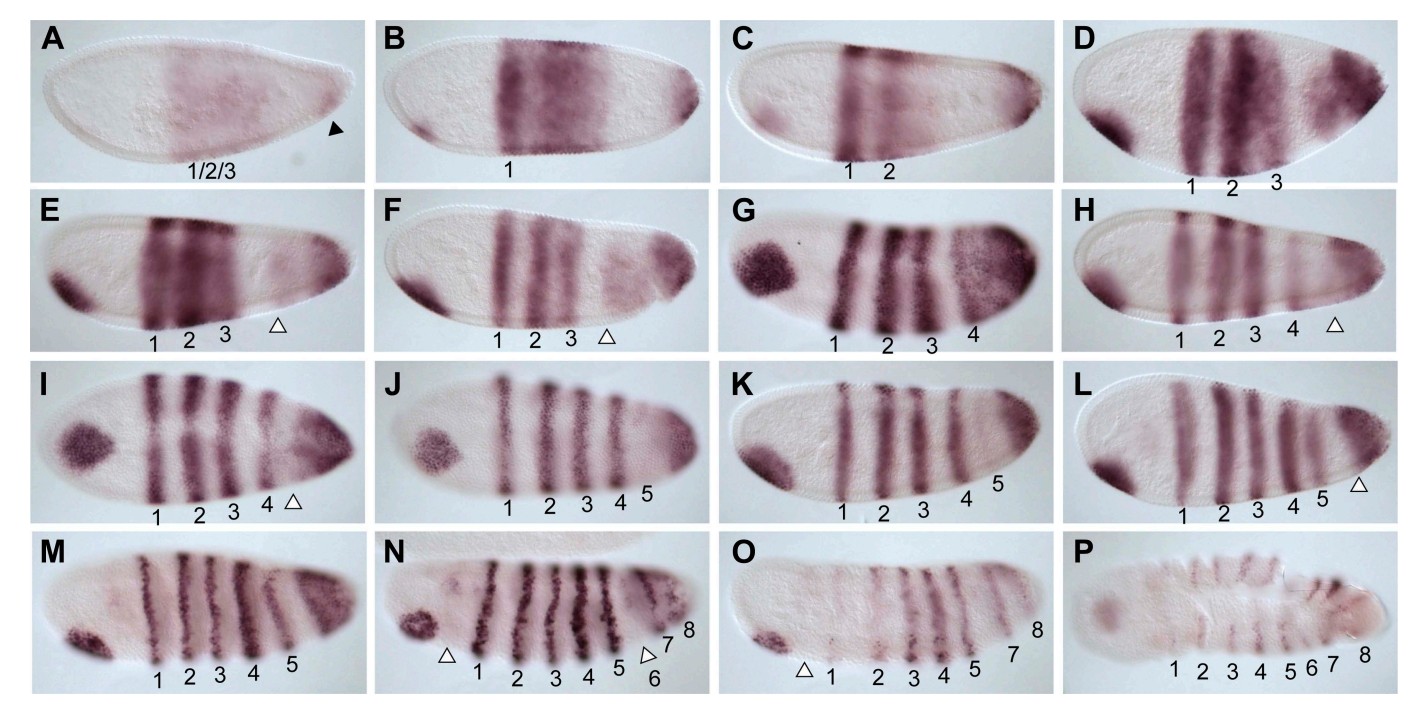

**Figure 5**. Summary of *Nv odd-skipped* mRNA expression. Embryos are shown with anterior left and dorsal up, except where indicated. (**A**) Precellular blastoderm embryo showing early expression of *Nv odd* in a broad domain and a posterior cap with a slight clearing in between. (**B**) Precellular blastoderm embryo showing ventral head patch and darkened central broad domain and distinct posterior cap. (**C**) Precellular blastoderm embryo with sharpening pair-rule stripes and expanding posterior cap. (**D**) Precellular blastoderm embryo with dark ventral head patch and posterior cap, and expansion of expression between broad central domain and posterior domain. (**E** and **F**) Cellularizing blastoderm embryos with three double-segment periodicity stripes, and a continuous posterior domain of variable staining intensity. Arrowhead indicates boundary of faint expression, which prefigures position of double-segment stripe 4. (**G**) Ventral view of cellularizing embryo with three strong double-segment stripes, and a fourth stripe forming at the anterior boundary of a more uniformly staining posterior cap (arrowhead). (**H**) Cellularized blastoderm embryo with four distinct double-segment stripes and a receding posterior cap domain (arrowhead). (**I**) Ventral view of cellular blastoderm showing four strong double-segment stripes and receding posterior cap (arrowhead), whose anterior boundary prefigures the position of stripe 5. (**J**) Ventrolateral view of cellular blasoderm embryo showing early appearance of stripe 5 at the anterior boundary of receding posterior domain, whose staining intensity is now less uniform. (**K**) Cellular blastoderm embryo with five double-segment stripes of expression, a strong ventral head spot, and a reduced, uniform posterior cap. (**L**) Same as **K**, with five equivalently strong double-segment stripes. Arrowhead indicates slightly expanded posterior cap. (**M**) Early germ-band extension embryo with five double-segment periodicity stripes and two stripes becoming evident within the posterior cap. (**N**) Slightly later embryo than **M**, with 2 posterior cap stripes more clearly differentiated. (**O**) Slightly later embryo than **N**, with anterior stripes fading and posterior segments expanding. (**P**) Dorsal view, dorsal closure embryo exhibiting eight single-segment periodicity stripes.

The following figure supplements are available for figure 5:

**Figure supplement 1**. Phylogenetic analysis of odd-skipped.

**Figure supplement 2**. odd-skipped protein sequence alignment.

double-segment stripes (PR stripes 4–6) then form sequentially, apparently as 'waves' of *Nv odd* expression, resembling the clock-driven stripes of *Tc'odd*. Finally, two *Nv odd* stripes form from a posterior cap. *Nv odd* knockdown affects anterior thoracic segments with a partial pair-rule phenotype; it also leads to the loss of posterior segments A5–A10.

## *Nasonia runt* expression

The complementarity between *Nv eve* and *Nv odd* is suggestive of cross interaction between the two genes but it is only partial and only affects half of the *Nv eve* segmental stripes since *Nv odd* does not have single-segment periodicity stripes. We sought to determine whether the remaining single segment stripes where *odd* is not interdigitated with *Nv eve* may alternate with stripes of *Nv runt*, as is observed

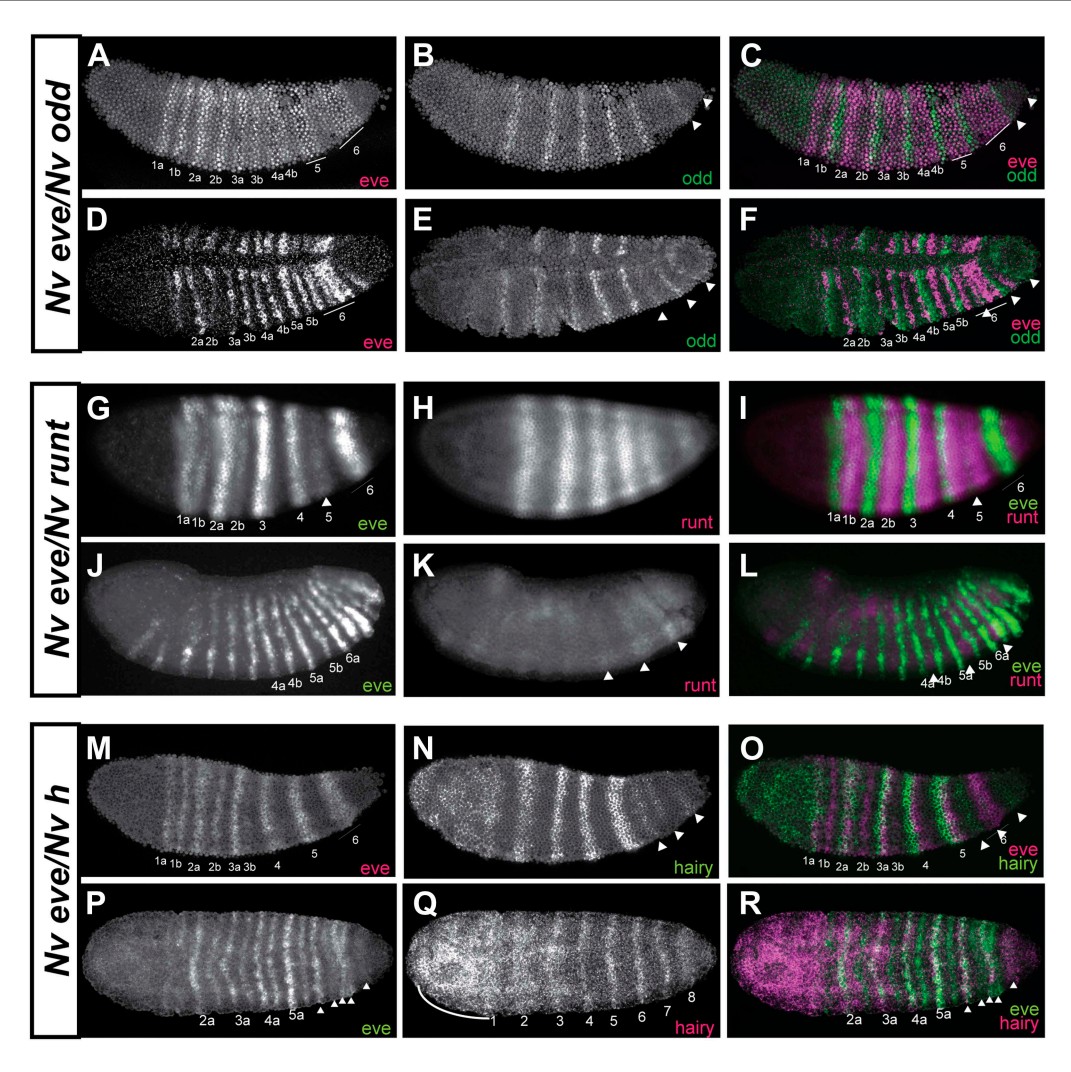

**Figure 6**. Phasing of Nasonia pair-rule genes in embryos using double fluorescent in situ hybridization. (**A**) Lateral view of *Nv eve* expression in early gastrula embryo. (**B**) *Nv odd* expression alone in the same embryo. (**C**) Merge of *Nv eve* and *Nv odd* channels, illustrating their relative phasing. *Nv eve* mRNA is pseudo-colored pink, *Nv odd* is in green. Arrowheads indicate position of a posterior doublet of *odd* stripes. (**D**) Dorsolateral view of *Nv eve* in later gastrula embryo. (**E**) *Nv odd* expression alone in the same embryo. Arrowheads indicate position of posterior *Nv odd* stripes 6, 7 and 8. (**F**) Merge of *Nv eve* and *Nv odd* channels, illustrating their relative phasing. (**G**) Lateral view of *Nv eve* expression in blastoderm embryo. Arrowhead indicates position of *Nv eve* stripe 5. (**H**) *Nv runt* expression in the same blastoderm embryo. (**I**) Merge of *Nv eve* (green) and *Nv runt* (pink) channels, indicating relative phasing. (**J**) Lateral view of *Nv eve* expression in germ-band-extended embryo. Numbers indicate identity of *Nv eve* stripe. (**K**) *Nv runt* expression alone in the same embryo. Arrowheads indicate position of posterior primary *Nv runt* stripes. (**L**) Merge of *Nv eve* (green) and *Nv runt* (pink) channels, indicating relative phasing. Note that posterior *Nv runt* stripes, though faint, appear to be positioned posterior to odd-numbered *Nv eve* segmental stripes. (**M**) Lateral view of *Nv eve* expression in early gastrula embryo. Line indicates broadening stripe 6. (**N**) *Nv hairy* expression in the same gastrula embryo. Arrowheads indicate positions of three late forming posterior double-segment stripes. (**O**) Merge of *Nv eve* (pink) and *Nv hairy* (green) channels, indicating relative phasing. (**P**) Ventral view of gastrula embryo showing *Nv eve* expression alone. Arrowheads indicate positions of single-segment stripes derived from *eve* stripe 6. (**Q**) *Nv hairy* expression alone in the same gastrula embryo. Line indicates extended anterior domain continuous with stripe 1. (**R**) Merge of *Nv eve* (green) and *Nv hairy* (pink) channels, illustrating relative phasing.

The following figure supplements are available for figure 6:

**Figure supplement 1**. Summary of *Nv runt* mRNA expression.

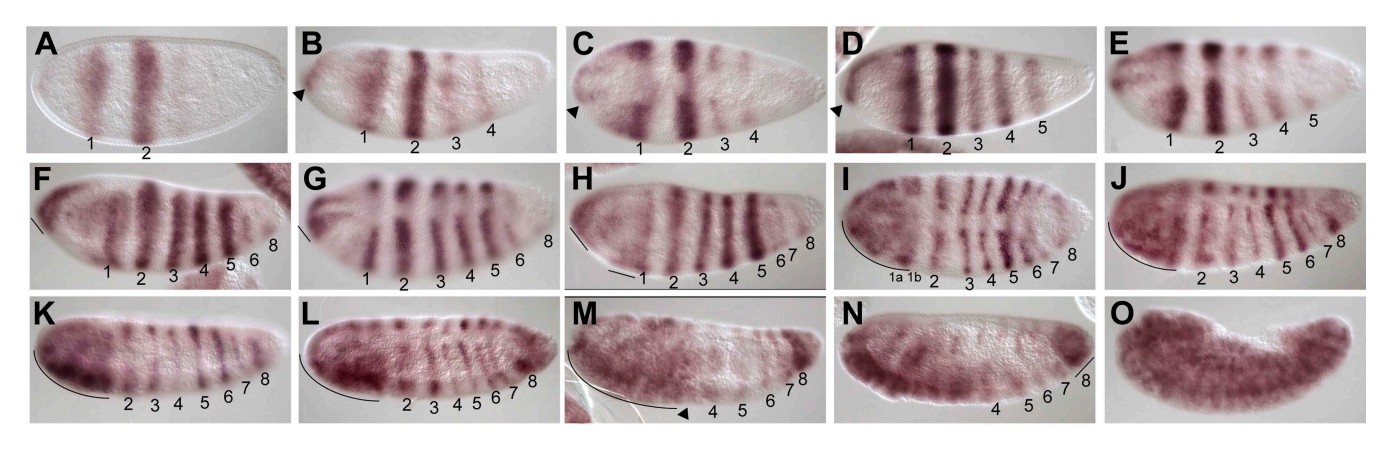

**Figure 7**. Summary of *Nv hairy* mRNA expression. (**A**) Blastoderm embryo with two double-segment periodicity stripes of *Nv hairy* expression. Note that stripe 2 is broader and stronger than stripe 1. (**B**) Blastoderm embryo showing four double-segment periodicity stripes of expression plus an anterior accumulation of *Nv hairy* transcripts (arrowhead). (**C**) Dorsal view of embryo as in (**B**), illustrating the dorsal anterior expression (arrowhead) that is activated in the same pattern as the anterior domain of *Nv tailless* (**Lynch et al., 2006**). (**D**) Blastoderm embryo with strong anterior and dorsal anterior expression of *Nv hairy* and five pair-rule stripes. (**E**) Dorsal view of embryo as in (**D**) with increased dorsal anterior expression of *Nv hairy*, and the anterior spreading of expression from the anterior of double-segment pair-rule stripe 1. (**F**) Blastoderm embryo with expanding anterior domain (line), five double-segment 'pair-rule' stripes, and two additional stripes coming up. Note that the anterior domain between stripe 1 and the anterior pole is becoming more continuous in expression. (**G**) Dorsolateral view of embryo as in (**F**) highlighting the dorsal anterior expression. Stripe 2 is still wider than other stripes. Stripe 6 appears to be of single-segment periodicity. (**H**) Early gastrula embryo exhibiting a non-homogenous but largely continuous anterior cap of *Nv hairy* expression (that includes stripe 1). Four additional double-segment stripes and three single-segment stripes (two derived from stripe 6) are now evident. (**I**) Dorsal view of embryo slightly older than embryo in (**H**) showing the nearly continuous head domain, and the apparent splitting of stripe 1 within that domain. Double-segment stripes are thinning. (**J**) Dorsolateral view of extending germ-band embryo. Head domain is continuous (line). Stripes 1–7 have single-segment periodicity, are of non-uniform strength; stripe eight appears darker and broader. (**K**) Germ-band extending embryo with a continuous head domain (line) and eight discrete stripes. (**L**) Dorsolateral view of germ-band extending embryo. Stripe 8 is expanded into a wedge abutting the pole cells, and the anterior domain is expanding to include stripe 2. (**M**) Germ-band extension embryo with expanding anterior domain, that extends to include stripe 3 (arrowhead). Posterior domain is expanded. (**N**) Dorsolateral view of embryo as in (**M**) showing further expansion of posterior stripe 8 domain (line). (**O**) Germ-band-retracted embryo exhibiting ubiquitous staining with striated expression evident.

The following figure supplements are available for figure 7:

**Figure supplement 1**. Phylogenetic analysis of hairy.

**Figure supplement 2**. Hairy protein sequence alignment.

in *Drosophila*. We studied the expression of *Nv runt* throughout embryogenesis (*Figure 6—figure supplement 1*) and then used double fluorescent in situ hybridization to visualize its register with *Nv eve* stripes at both early and late stages. *Nv runt* stripes appear cleanly in an anterior to posterior progression, with six double-segment periodicity stripes visible before cellularization; two additional double-segment stripes are added at the posterior during gastrulation. Single-segment periodicity stripes only appear much later at full germ band extension when the expression of the other pair-rule genes is already well established (*Figure 6—figure supplement 1*). In the early embryo, *Nv runt* double-segment stripes appear posterior to, and partly overlapping with, each *Nv eve* primary double-segment stripe (*Figure 6G–I*). Splitting of anterior *Nv eve* stripes moves the posterior of each doublet (i.e., even-numbered *Nv eve* single-segment stripes) more posteriorly beyond each *Nv runt* primary double-segment stripe (*Figure 6G–I*). The appearance of *Nv eve* stripe 5 between *Nv runt* double-segment stripes 4 and 5 as they split (*Figure 6G–I*, arrowhead; *Figure 6—figure supplement 1F,G*) suggests that *eve* may help to repress *Nv runt*, though this remains to be tested.

Late expression of *Nv runt* in the extending germ band is considerably weaker than that of other genes, making its detection more challenging. Still, at the posterior of the embryo, *Nv runt* double-segment stripes appear between *Nv eve* single-segment stripes arising from splitting of double-segment stripes (*Figure 6J–L*). Altogether, our data support a model in which *Nv eve* single-segment

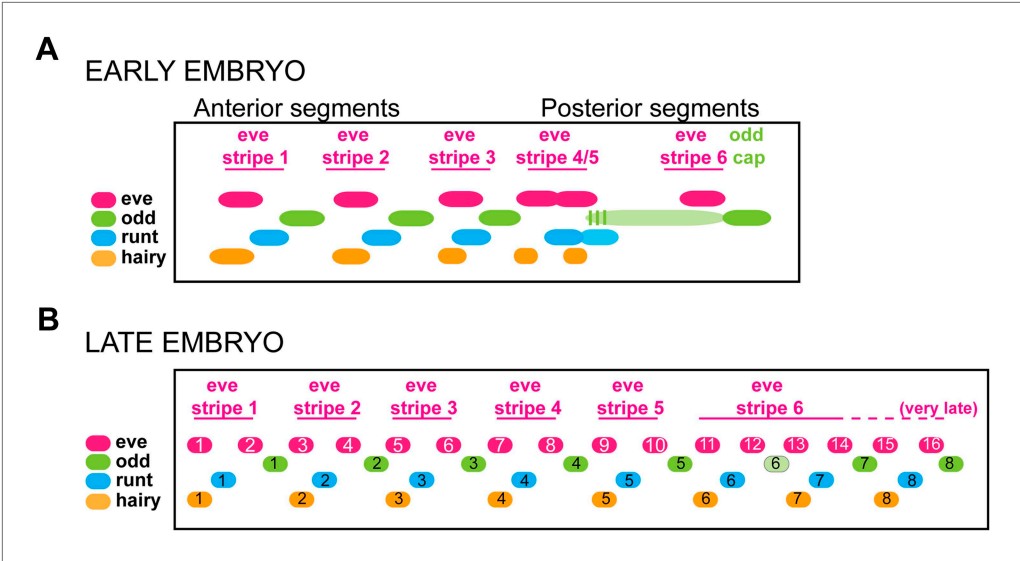

**Figure 8**. Summary model of pair-rule gene expression in the Nasonia embryo. (**A**) Model of register of pair-rule gene expression in the early embryo. *Nv eve* and *Nv odd* stripes are totally complementary, whereas *Nv runt* stripes partly overlap each of these genes at their interface. *Nv hairy* stripes overlap *Nv eve* stripes toward the anterior of each double-segment periodicity stripe. Towards the posterior of the embryo, an extended domain of low-level *Nv odd* expression exhibits dynamic behavior over several nuclear cycles, and stripes 4/5 of *Nv eve, Nv odd*, and *Nv runt* each differentiate during this interval. Even more posteriorly, *Nv eve* stripe 6 lay anterior to a continuous *Nv odd* cap that extends to the posterior pole of the embryo. This region is set aside for segment specification and differentiation during germ-band extension. (**B**) Model of register of pair-rule gene expression in the germ-band extension (late) embryo. Single-segment periodicity stripes in the germ-band-extended embryo exhibit a variation upon early gene expression patterns. *Nv eve* single-segment stripes are interrupted by *Nv runt* and then *Nv odd* such that *Nv runt* stripes follow odd-numbered *Nv eve* stripes, and *Nv odd* stripes follow even-numbered *Nv eve* stripes. Each of 8 *Nv hairy* stripes overlaps odd-numbered *Nv eve* stripes that derive from the anterior of *Nv eve* pair-rule stripes. Additional expression of several of these genes in the ventral and head domains, which appears to rely on different regulatory logic, is not shown.

periodicity stripes are established through the complementary action of *Nv odd* for odd numbered single-segment stripes and *Nv runt* for even-numbered stripes, as summarized in our model below. However, these interactions are still speculative since we have been technically unable to complete the epistasis experiments needed to test this model. A predicted role for *Nv h* is also described below.

## Expression and function of *Nasonia hairy*

*hairy* is a primary pair-rule gene in *Drosophila*, but in *Tribolium*, its function is restricted to head segment differentiation (*Carroll et al., 1988*; *Carroll and Vavra, 1989*; *Edgar et al., 1989*; *Choe et al., 2006*). There are two *hairy*-like genes in *Nasonia*, and we identified the likely *hairy* (*h*) ortholog using phylogenetic analysis (*Figure 7—figure supplements 1 and 2*). We examined expression of *Nv h* using a probe directed against the full-length coding region (Genbank Accession # KC190514).

*Nv h* expression begins as a single broad anterior double segment stripe 1 that incompletely spans the dorso–ventral axis. It is soon followed by a second broad double-segment stripe 2 just anterior to the middle of the embryo (*Figure 7A*). Double-segment stripes 3, 4 and 5 are then added sequentially before cellularization (*Figure 7B–E*); a faint stripe at the extreme posterior of the embryo is also visible. By gastrulation, an anterior cap becomes more visible with continuous low expression between the anterior pole and the strong stripe 1 (*Figure 7H–J*). As gastrulation progresses, this domain becomes stronger and more uniform, whereas double-segment stripes 6 and 7 appear sequentially (*Figure 7H–K*). Stripe 8 broadens, becoming a posterior cap whose intensity increases during germ band extension (*Figure 7J–N*). The anterior of the embryo exhibits diffuse staining that expands from the anterior ventral side posteriorly, until germ band retracted embryos are faintly but uniformly stained with dark segmental stripes on top (*Figure 7O*). *Nv h* double-segment stripes appear cleanly, and the timing and

presentation of expression of posterior stripes suggest that they may respond to waves of *Nv odd*. Like *Nv odd* and early *Nv runt*, *Nv h* does not have stripes with single-segment periodicity.

Double fluorescent in situ hybridization with *Nv eve* reveals that early *Nv h* overlaps the anterior of *Nv eve* double-segment stripes (***Figure 6M–O***). Later, these stripes appear thinner, and they overlap the anterior *Nv eve* single-segment odd-numbered stripes in each double-segment doublet. *Nv h* double-segment stripe 6 anticipates the position of the most anterior derivative of the *Nv eve* stripe 6 quartet (segmental stripes 11–14). *Nv h* stripe 7 (double segment), which is thin, appears within the *Nv eve* early broad stripe 6 domain, coinciding with *Nv eve* single-segment stripe 13 (***Figure 6P–R***, ***Figure 8***). A more posterior stripe, *Nv h* 8, anticipates, albeit more broadly, the site of *Nv eve* single-segment stripe 15 (***Figure 6M–O,P–R***, ***Figure 8***). Thus, *Nv h* and *Nv eve* are co-expressed at the anterior of *eve* pair-rule stripes and in the first of each pair of *eve* (odd-numbered) single-segment stripes, similar to the relationship described for *Drosophila eve* and *hairy* as they initiate segment polarity (***Warrior and Levine, 1990***).

We knocked down *Nv hairy* function using two independent morpholinos directed at two different splice junctions. Both resulted in a range of cuticle defects that indicate that *Nv h* is required for the formation of all posterior-derived segments and blastoderm-derived segments in the thorax and anterior abdomen (***Figure 4L–O***). At the posterior, mildly affected cuticles exhibit fusion of segments A9–10 (***Figure 4L,N***), along with partial loss of alternating abdominal segments posterior to A4. In more affected cuticles, alternating segments posterior to A2 are fused (***Figure 4L–N***), resembling a pair-rule phenotype. In severely affected embryos, all segments from A4–A10 are fused with a continuous lawn of denticles that covers the posterior of a severely reduced cuticle (***Figure 4O***, yellow line). These phenotypes suggest a requirement for *Nv hairy* in specification of posterior segments. The late *Nv h* stripes 6–8 are positioned to affect the late forming segments as supported by double *in situs* (***Figure 6***). In spite of *Nv hairy* expression in the head and extreme anterior of the embryo, labral structures in *Nv hairy* morpholino cuticles appear to be unaffected. The expression pattern of *Nv hairy* is thus strikingly similar to *Tc'hairy* (***Sommer and Tautz, 1993***; ***Aranda et al., 2008***), though functionally quite different, since *Tc'h* seems to act exclusively in patterning head segments (***Choe et al., 2006***; ***Aranda et al., 2008***).

At the anterior, A2 is also nearly always affected, exhibiting loss of denticles and displacement or loss of the associated spiracle (***Figure 4L***, orange arrowhead). Segment T1 appears to be lost and fused to T2. Finally, more affected embryos show a loss of T3 (***Figure 4M–O***). Therefore, segments T1, T3, and A2 are missing, which resembles an anterior pair-rule phenotype.

In summary, *Nv h* expression is highly dynamic and proceeds in an anterior to posterior progression. It is distinct from *Nv eve* and does not exhibit single-segment periodicity stripes. At the end of embryogenesis, its expression becomes nearly ubiquitous (***Figure 7O***).

Taken together, these data support a model wherein 'pair-rule' genes have weak fly-type 'pair-rule' phenotypes in the anterior, and are required for the formation of a suite of posterior segments. Their interdigitated expression suggests extensive interactions during patterning of the posterior region after cellularization, although this has not yet been tested due to current experimental limitations. Our summary model of the phasing of 'pair-rule' stripes in the embryo is given in ***Figure 8***.

## Discussion

Decades of study of a variety of insects have yielded a deep understanding of the genes controlling anterior–posterior patterning of the embryo. The best-characterized model species are the extreme long germ *Drosophila* embryo and the short-germ *Tribolium* embryo. Additional insects, such as *Gryllus bimaculatus* (***Mito et al., 2007***), *Bombyx mori* (***Xu et al., 1997***), *Oncopeltus fasciatus* (***Liu and Kaufman, 2004a***, ***2004b***, ***2005a***), *Schistocerca americana* (***Patel et al., 1992***) and others, represent informative intermediates, but many of these species are on the short end of this wide spectrum, consistent with ancestral insects being short germ band. *Hymenoptera*, including *Apis* and *Nasonia*, have evolved a long germ embryogenesis independently from flies and therefore provide an excellent context for addressing the question of how the transition from short to long germ occurred. Other well-characterized arthropod species, such as *Cupiennius salei* (***Damen et al., 2000***; ***Stollewerk et al., 2003***) and *Strigamia maritima* (***Chipman et al., 2004***; ***Chipman and Akam, 2008***), provide additional models of interest for understanding more ancient evolutionary history.

*Nasonia* has been characterized as long germ because of the presence of two morphogenetic centers and the expression patterns and RNAi phenotypes of the gap genes (***Ingham et al., 1985***; ***Pultz et al., 2005***; ***Lynch et al., 2006***; ***Olesnicky et al., 2006***; ***Brent et al., 2007***). In this study, we describe the

expression and loss of function phenotypes of *Nasonia eve*, *hairy* and *odd*, three genes that act as pair-rule genes in *Drosophila.* While the expression of *Nv eve* is controlled by maternal and gap gene factors that are largely similar to their *Drosophila* counterparts, we find critical deviations from the *Drosophila* long germ paradigm. *Nv* 'pair-rule' genes display similarity to both *Drosophila* and *Tribolium,* suggesting that *Nasonia* has features of both short and long germ band development. Unlike the long germ embryo of the honeybee, *Apis*, *Nasonia* pair-rule genes do not seem to be maternally expressed, and so far, regulation of *Nv* gap genes by pair-rule genes (as reported for *Apis* [**Wilson and Dearden, 2012**]) has yet to be studied systematically.

## Expression and knockdown of *Nasonia eve*, *hairy* and *odd*

In contrast to *Drosophila,* whose pair-rule genes are expressed in the blastoderm in seven double-segment periodicity stripes to determine the formation of 14 segments, their orthologs in *Nasonia* are expressed in diverse and more intricate patterns. In no case do we observe simply eight precellularization double-segment stripes, confirming that pair 'rule' does not represent the regulatory dynamics of these genes across insects. We observe wave-like behavior of *Nv odd* stripe 4–6, which underscores that cycling control may remain from the ancestral segmentation clock. *Nv runt* and, to a large degree, *Nv h*, also exhibit a sequential progression of sharp stripe appearance that may be responsive to the waves of *Nv odd* (*Figure 7*, *Figure 6—figure supplement 1*).

It is clear that *Nv eve, Nv odd,* and *Nv h* exhibit multiple modes of regulation during embryogenesis. In each case, their anterior stripes formed in the syncytial blastoderm have double-segment periodicity and arise in a manner that could be explained by the type of enhancer logic exemplified by *Drosophila eve* (*Small and Levine, 1991*; *Small et al., 1992*, *1996*; *Schroeder et al., 2004*; *Schroeder et al., 2011*). Anterior double-segment 'pair-rule' stripes of *Nv eve* appear to be regulated by maternal and gap genes as in *Drosophila,* and embryos knocked down for *Nv eve, Nv odd* and *Nv h* exhibit a pair-rule phenotype in the blastoderm-derived segments, although this phenotype is most often limited. It is worth noting that the severe *Nv eve* anterior defects are more regional, as observed for *Oncopeltus eve* that behaves as a gap gene, and is attributed to the broad early domain of expression (*Liu and Kaufman, 2005a*). Perhaps, as in *Oncopeltus, Nv eve* is required in combination with *Nv hb* or *Nv gt* for activation of their targets, which are in turn required for the proper formation of head and thoracic segments.

Another mode must control the formation of stripes of *Nv eve, Nv odd,* and *Nv h* that arise later, in a cellular environment, from a posterior domain (whether at the posterior pole of the embryo, as in the case of *Nv odd* and *Nv hairy*, or from a broad posterior stripe, as in the case of *Nv eve*). Knockdown of each of the three genes produces a severe posterior truncation of the embryo, deleting all six posterior segments, indicating that each gene is required for the formation of posterior-derived segments. This phenotype is unlike flies, and resembles more the short germ pair-rule gene circuit of *Tribolium* (*Choe et al., 2006*; *Choe and Brown, 2009*; *Sarrazin et al., 2012*). Together with co-expression data, their phenotypes suggest that each of these genes is required for refinement or maintenance of each other's activity or expression (*Figure 6*, *Figure 6—figure supplement 1*).

We propose a model in which interactions among 'pair-rule' genes dominate in patterning the long germ *Nasonia* embryo. Unlike flies, posterior stripes of 'pair-rule' genes like *Nv hairy* and *Nv runt* appear sequentially. Indeed, the gene circuit involving interactions among *Tc'odd, Tc'eve* and *Tc'runt* in each round of posterior segment formation underscores the likely ancestral nature of this network, which might have been brought under the control of the gap and maternal genes in flies, and in the anterior segments of *Nasonia*. The 'waves' of *Nv odd* pair-rule stripe expression that give rise to blastoderm stripes 4, 5, and 6 suggest residual activity of a segmentation clock. The presumptive domain of six posterior segments indicated by early posterior expression of *Nv eve* and *Nv odd* may be similar to the 'growth zone' of short germ insects.

Waves of *Nv odd* 4, 5, and 6, and the sequential formation of *Nv runt* stripes both interrupt and likely pattern the continuous *Nv eve* stripe 6 domain (*Figure 6A–L*; *Figure 8*). *Nv h* expression anticipates the final position of several late forming *Nv eve* stripes, and in combination with the phenotype of *Nv h* knockdown and co-expression data, suggests that it is required for the formation of the posterior *Nv eve* stripes.

Thus, *Nasonia* represents a variation on embryo allocation and patterning, but the contribution of 'pair-rule' gene function is enduring. The use of a clock-like mechanism is not incompatible with long germ embryogenesis, and rather, retaining this character might allow for sampling transitional states

between the short- and long germ strategies, which likely occurred multiple times within holometabola. Further, it may be that the absence of significant posterior elongation is the transition state that tips the balance toward elaboration of anterior segmentation control mechanisms and loss of late forming segments. Our characterization of the *Nasonia* pair-rule genes illustrates one way that these two strategies can co-exist.

It is also of note that although *hairy*-related genes are the oscillating components of vertebrate segmentation clocks (*Palmeirim et al., 1997*), it is generally *odd-skipped*-related genes that oscillate in arthropods (*Chipman et al., 2004*; *Chipman and Akam, 2008*; *El-Sherif et al., 2012*; *Sarrazin et al., 2012*). Notch-signaling has been described for its involvement in regulating *hairy*-related oscillations in the vertebrate clock (e.g., *Jouve et al., 2000*), and it may also be involved in the context of the arthropod segmentation clock (*Stollewerk et al., 2003*; *Eriksson et al., 2013*; *Kainz et al., 2011*); in all cases, the driver of the clock is yet to be elucidated (reviewed in *Pourquie, 2003*).

### An ancestral role for *eve* in specifying posterior identity may be linked to growth zone behavior in short germ insects

*eve* has been suggested to have its most ancestral function as a specifier of posteriorness (*Ahringer, 1996*; *Brown et al., 1997*). Indeed, the two mammalian *eve* genes are located at the most 'posterior' end of two of the Hox clusters (*Bastian et al., 1992*), although *eve* is not a part of the Hox cluster in *Nasonia* or *Tribolium* or any other insects that have been studied (*Shippy et al., 2008*; *Werren et al., 2010*; *Munoz-Torres, 2009*; *Suen et al., 2011*). Yet, even in *Drosophila*, where there is no apparent sequential segmentation, a delayed pair-rule stripe (stripe 8) appears early in gastrulation (*Macdonald et al., 1986*; *Frasch et al., 1987*; *Kim et al., 2000*). In *Schistocerca*, *eve* is expressed in a posterior mesodermal domain and no pair-rule stripes arise from this region, indicating that *eve* plays a role in posteriorness and not segmentation in basal insects (*Patel et al., 1992*). *Nasonia eve* sets aside stripe 6 relatively early, at about the same time as *Nv odd* that is expressed even more posteriorly. This feature of *Nv eve* in posterior segmentation is not shared by the other pair-rule genes we studied, therefore supporting the notion of an ancestral role for *eve* in posteriorness in *Nasonia*. That its expression is complementary to that of *odd* in both *Tribolium* and *Nasonia* in a late-differentiating posterior region may hint at how this role in posteriorness evolved into a role in posterior growth. In non-insect arthropods, there is evidence for a role for *eve* in both posterior identity and segmentation. The centipede *Lithobius atkinsoni* expresses *eve* in a posterior domain and between segments (*Hughes and Kaufman, 2002*), and the crustacean *Artemia franciscana* exhibits growth zone *eve* expression that precedes expression in stripes in emerging segments (*Copf et al., 2003*). In other basal arthropods, like spiders (*Damen et al., 2000*) and the centipede *Strigamia maritima* (*Chipman and Akam, 2008*), *eve* expression in stripes suggests that its ancestral role is segmental.

### Mitotic domains are coordinated with pair-rule gene expression

The broad stripe 6 domain of *eve* appears to be subdivided by transcription control, likely through interactions with *Nv odd* and *Nv runt* (*Figure 6*, *Figure 6—figure supplement 1*). Although we observed apparent mitotic domains in the early gastrula that proceed from anterior to posterior, they do not match the pattern of initial differentiation of germ band-derived segments or the splitting of anterior pair-rule stripes. Cell division patterns in mitotic domains have not been described in most insects, apart from *Drosophila* and the precellular blastoderm of *Bombyx* (order: Lepidoptera; *Nagy et al., 1994*). In the short germ *Tribolium* and *Oncopeltus* embryos, cell divisions during gastrulation and elongation occur throughout the germ band, with no evidence for mitotic domains (*Brown et al., 1994*; *Liu and Kaufman, 2009*).

The relationship between *Nv eve* and cell division suggests coordination of cell divisions by segmentation genes, a phenomenon that has been suggested for *Drosophila* (*Foe, 1989*; *Edgar and O'Farrell, 1989*; *Bianchi-Frias et al., 2004*). Use of coordinated mitotic domains is a strategy that seems to have evolved multiple times (e.g., in flies and wasps). We propose that the apparent coordination of mitotic domains and segmentation gene expression of both *Nasonia* and *Drosophila* development may constitute a step in the transition to long germ embryogenesis.

### Conclusion

In summary, despite obvious differences in their expression patterns, *Nasonia eve*, *odd*, and *hairy* function in both blastoderm- and germ band-derived segment formation. While *Nasonia* exhibits

fly-type expression of maternal and gap genes in the precellular blastoderm, dynamic expression patterns and extensive interactions among 'pair-rule' genes appear to pattern a suite of late forming posterior segments. Indeed, their relative expression patterns suggest that the regulation of posterior segments may be through the type of mutual regulation described for the pair-rule gene circuit of *Tribolium*. This is unlike the long germ embryo of *Drosophila,* whose segmentation utilizes pair-rule interactions only during the late blastoderm stage. We propose that late-forming segments are set aside using remnants of ancestral control of posteriorness and the segmentation clock. Thus, *Nasonia* relies on a dynamic, dual mode of segmentation that has characteristics of both ancestral short germ and derived long germ embryogenesis.

## Materials and methods

### Embryo collection, non-fluorescent in situ hybridization, two-color FISH, and immunohisochemistry

*Nasonia* embryos were collected and fixed in 5% formaldehyde/1X PBS/Heptane for 28 min, affixed to double-sided tape, and hand peeled under 1X PBS +0.1% tween, as described previously (*Pultz et al., 2005*), except that the embryos were collected from host-fed, mated females. The embryos were stored under methanol at −20°C between fixation and hybridization.

In situ hybridizations were carried out as described previously (*Pultz et al., 2005*). Briefly, the embryos stored under methanol were gradually brought up to 1X PBT and washed three times in 1x PBS +0.1% tween-20 (PBT) before a 30-min post-fix step in 5% formaldehyde/1XPBT. The embryos were then washed three times and subjected to proteinase K treatment (final concentration of 4 µg/ml) before three PBT washes. The embryos were blocked for 1 hr in hybridization buffer before probe preparation and addition for overnight incubation at 65°C. The next day, the embryos were washed in formamide wash buffer and then 1X MABT buffer before blocking in 2% Blocking Reagent (BBR; Roche Applied Science, Germany) in 1X MABT for 1 hr, then in 10% horse serum/2% BBR/1XMABT for 2 hr. The embryos were incubated overnight with primary antibody in the second blocking solution at 4°C. Anti-DIG-AP Fab fragments (Roche Diagnostics) were used at 1:2000 for non-fluorescent in situs. On the third day, the embryos were washed in 1X MABT for ten, 20 min washes before equilibrating the embryos in AP staining buffer and developing in AP buffer with NBT/BCIP solution (Roche Diagnostics). After staining, the embryos were washed in 1× PBT three times for 5 min each before a single 25 min post-fix step in 5% formaldehyde/1XPBT. The embryos were then washed briefly and allowed to sink in 50% and then 70% glycerol/1XPBS, which were subsequently used for mounting.

For fluorescent in situs, DIG probes were detected using anti-DIG-POD Fab fragments (Roche Diagnostics) at 1:50 dilution, followed by FastRed HNPP detection system (Roche Diagnostics), according to manufacturer's instructions. Fluorescein probes were detected using anti-Fluorescein-AP Fab fragments (Roche Diagnostics) at 1:500 dilution.

For antibody staining of mitotic cells, we used a rabbit anti-phosphorylated histone H3 serine 10 antibody (Millipore, Billerica, Massachusetts) at 1:200, and then a donkey anti-rabbit secondary conjugated to Alexa-647 (Invitrogen, Carlsbad, California) at 1:200. In combination with FastRed in situ detection using anti-DIG-AP Fab fragments (Roche Diagnostics) at 1:500, primary antibodies were added to blocking buffer together and incubated according to the in situ protocol, and secondary antibody detection was carried out after the FastRed staining was completed and following three 1X PBT washes.

### Cloning of *Nasonia* pair-rule gene cDNA fragments

*Nasonia* pair-rule gene cDNAs were cloned from embryo cDNA pools generated from reverse transcription of total embryo RNA from mixed age embryos using Superscript First Strand Synthesis kit (with Superscript II; Invitrogen) according to manufacturer's specifications. For cases in which long cDNA sequences could not be amplified with oligos designed according to automated genome annotation and prediction models, we used circular RACE to simultaneously amplify sequences 5′ and 3′ to smaller cloned cDNA fragments, as previously described (*McGrath, 2011*).

### Phylogenetic analysis of *Nasonia* pair-rule gene paralogs

*Nasonia* paralogs of fly pair-rule genes were identified by TBLASTN and aligned against predicted or experimentally validated (virtually translated) protein sequences of the same proteins from *Tribolium*

castaneum (*Tcas*), *Anopheles gambiae* (*Agam*), *Apis mellifera* (*Amel*). Protein sequences were aligned using CLUSTALW (*Larkin et al., 2007*) and rendered using Dendroscope (*Huson et al., 2007*). Evolutionary relationships were inferred using a maximum likelihood analysis with 1000-fold bootstrap support, via RaXML hosted online at CIPRES science gateway (http://www.phylo.org/index.php/portal/) (*Stamatakis et al., 2008*; *Stamatakis, 2006*; *Miller et al., 2010*).

## Morpholino injection and larval cuticle preparations

Antisense morpholinos targeting splice junctions or transcription initiation sites were designed and ordered from GeneTools, LLC (www.gene-tools.com, Philomath, Oregon). Lyophilized morpholinos were resuspended in sterile nuclease-free water to a final concentration of 5 mM. For *Nv odd* splice block morpholino, which yielded a high percentage of dead embryos with no cuticle, injections were also carried out at 1 mM, 0.5 mM, and 0.05 mM dilutions. *Nasonia* embryos were collected for 35 min at 28°C and dehydrated for 30 min before injection with morpholinos (approximate volume injected = 0.001 µl per embryo). The embryos were allowed to develop on the injection membrane at 28°C on a 1X PBS/1% agarose plate for approximately 30 hr, to ensure complete development. Unhatched larvae were peeled and transferred to a slide for cuticle preparations in Lacto–Hoyer's media.

Morpholino sequences used are as follows:

Eve translation block 5' CAAAGCTCCTCTGGAATCCTTGCAT 3'
Eve E2I2 splice block: 5' AAACGATAGTTACCTTGATGGTCGA 3'
Hairy E2I2 splice block: 5' CTGAATCTGTCAAGATACTTACGTC 3'
Hairy E1I1 splice block: 5' GAGCAAGTCGAGATACTAACCCGTC
Odd splice block: 5' AGAGAGTGTACTAAC TTGTGGTCCC 3'
Odd translation block: 5' GCTCCATCGCAAGCTGGGTAAACGT 3'

## cDNA sequence accession numbers

*Nv odd* cDNA GenBank Accession # KC142194
*Nv eve* cDNA, isoform 1 GenBank Accession# KC168090
*Nv eve* cDNA, isoform 2 GenBank Accession# KC168091
*Nv eve* cDNA, isoform 3 GenBank Accession# KC168092
*Nv hairy* cDNA GenBank Accession # KC190514

Accession numbers for sequences used in sequence alignments and trees:

NvitH1: NP_001267498 XP_001601817 (Nvit Hairy)
NvitH2: Uniprot K7J0X7_NASVI (hairy-like/Nvit Dpn)
NvitH3: XP_001601600.2 GI:345484850 (hairy-like/HES like?)
DmelH: NP_523977.2 GI:24661088 (Dmel hairy)
DmelDpn: NP_476923.1 GI:17136808 (Dmel deadpan)
AmelH1: XP_001120814.2 GI:328784100 (hypothetical protein)
AmelH2: XP_393948.3 GI:110762302 (hairy-like)
AgambH1: XP_316733.3 GI:158296333 (corrected seq; Agam hairy)
AgambH2: XP_320206.4 GI:158300226
TcasH1: NP_001107765.1 GI:166796106 (Tcas Hairy)
TcasH2: XP_967694.1 GI:91092620 (Tcas similar to GA21268-PA)
TcasH3: XP_975187.1 GI:91083981 (Tcas HES1)
DmelOddsk: NP_722922.1 GI:24581484 (Dmel Odd skipped)
DmelSobow: NP_476882.1 GI:17136746 (Dmel Sister of odd and bowl)
DmelBowl: NP_476883.1 GI:17136748 (Dmel Brother of odd with entrails limited)
NvitOddbowlA: XP_001603713.1 GI:156545195 (predicted protein)
NvitOddbowlB: XP_001603827.2 GI:345481739(Nv bowel-like)
NvitOddbowlC: XP_001603660.1 GI:156545193(Nv odd-skipped like)
AmelOddbowlA: XP_001120949.1 GI:110762343(Amel odd-skipped like)
AmelOddbowlB: XP_393879.3 GI:110762378(Amel bowel-like)
AmelOddbowlC: XP_001120905.1 GI:110762341
TcasBowl: XP_972138.2 GI:189240088(Tcas bowl-like)
TcasOdd: XP_972086.2 GI:189240086 (Tcas odd-skipped)

TcasSob: XP_972035.1 GI:91088523(Tcas: predicted sister of odd and bowl)
Agam7972_PA: XP_306979.3 GI:118776890
Agam7973_PA: XP_317495.3 GI:118789549
Agam8222: XP_555242.1 GI:57914799

For *Figure 4*, the phenotypic classes are approximately as follows:

Eve: B 20.4% C 24% D 30.1% E 25%.
Odd: G 27.7% H 22.3% I 29.2% J 20.8%.
Hairy: L 13.4% M 25.0% N 36.6% O 25.0%.

## Acknowledgements

The authors would like to acknowledge Karin Kiontke and David Fitch for help with phylogenetic analyses, Jeremy Lynch for technical advice about multiple FISH, and Bruce Edgar and Pat O'Farrell for thoughtful comments and helpful advice on cell cycle data. Filipe Pinto Teixeira Sousa and Claire Bertet provided invaluable expertise and assistance with confocal imaging, and Jeremy Lynch and David Loehlin provided helpful comments on the manuscript. Terry Blackman and Cleo Tsanis provided immeasurable support, carrying out all of the injections described in the paper. The authors would also like to thank Bob Johnston, Leatt Gilboa, Zhenqing Chen, Michael Perry and Brent Wells for support and discussion during the course of this project. MIR would like to dedicate this paper in loving memory of Allen Rosenberg (1931-2013).

## Additional information

### Funding

| Funder | Grant reference number | Author |
|---|---|---|
| National Institutes of Health | F32GM084563 | Miriam I Rosenberg |
| Damon Runyon Cancer Research Foundation | DRG-1870-05 | Ava E Brent |
| American Cancer Society | 120323-PF-11-242-01-DDC | Miriam I Rosenberg |
| National Institutes of Health | 5R01GM064864 | Claude Desplan |

The funders had no role in study design, data collection and interpretation, or the decision to submit the work for publication.

### Author contributions

MIR, AEB, Conception and design, Acquisition of data, Analysis and interpretation of data, Drafting or revising the article, Contributed unpublished essential data or reagents; FP, Conception and design, Acquisition of data, Analysis and interpretation of data, Drafting or revising the article; CD, Conception and design, Analysis and interpretation of data, Drafting or revising the article

## Additional files

### Major datasets

The following previously published datasets were used:

| Author(s) | Year | Dataset title | Dataset ID and/or URL | Database, license, and accessibility information |
|---|---|---|---|---|
| | 2013 | UniprotKB data set for Nasonia vitripennis | http://www.uniprot.org/uniprot/?query=author:%22EnsemblMetazoa%22 | Publicly available at UniProt (http://www.uniprot.org/). |
| Werren JH, et al. | 2010 | Nasonia vitripennis Official Gene set | http://www.hymenopteragenome.org/nasonia/?q=sequencing_and_analysis_consortium_datasets | Publicly available at NasoniaBase (http://www.hymenopteragenome.org/nasonia/). |

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
