## [Decision Letter]

Thank you for sending your work entitled “Dual mode of embryonic development is highlighted by expression and function of *Nasonia* pair-rule genes” for consideration at *eLife*. Your article has been favorably evaluated by a Senior editor and 2 reviewers, one of whom, Duojia Pan, is a member of our Board of Reviewing Editors.

The Reviewing editor and the other reviewer discussed their comments before we reached this decision, and the Reviewing editor has assembled the following comments to help you prepare a revised submission.

Rosenberg et al. describe the expression patterns and phenotypes that result from knocking down several pair-rule genes in the wasp *Nasonia*. Previous studies, focused mostly on maternal and gap genes, had suggested that *Nasonia* uses a long-germ mode of embryogenesis, which evolved independently of long-germ development in flies. The results presented here lead the authors to re-evaluate that premise: pair-rule expression and phenotypes in *Nasonia* suggest both a 'long-germ' mode of action in anterior segments (patterning by subdivision in a syncytial blastoderm) and a 'short-germ' mode in posterior segments (sequential patterning in a cellularized environment, coupled with posterior elongation). The data are of excellent quality and largely support this interesting conclusion.

The reviewers suggested several major points for improvement:

1) The paper contains a number of imprecise statements that should be corrected by careful editing of the manuscript. On several occasions the authors refer to cross-regulatory interactions among the pair-rule genes, based entirely on expression patterns (e.g., several points in the Discussion section). I think that is a weak basis for proposing such interactions. I recommend testing these hypotheses by combining morpholino treatments with the relevant in situs. Alternatively, the authors should make it clear that the proposed interactions are hypothetical. The remaining suggestions are highlighted in the minor comments below.

2) The reviewers recommend separating, where possible, the description of data from the interpretations and the discussion that derives from it. Currently, much of the discussion is scattered within the Results section; in the Discussion the authors reiterate many of the same points.

Minor comments:

1) Introduction, 2^nd^ paragraph: I am not aware of evidence suggesting that *odd* controls the oscillations, rather than being a read-out?

2) Introduction, 2^nd^ paragraph: Whether the common ancestors of bilaterians was segmented is still controversial.

3) When describing the cuticular phenotypes induced by morpholinos (Figure 4 and Results section), it is not always clear how the authors identify which of the segments are missing or affected? Are there distinctive cuticular features that identify each segment? These should be described.

4) Section entitled ‘*Nasonia runt* expression’: “Altogether, our data support a model in which *Nv eve* segmental stripes are established through the complementary action of *Nv odd* for odd numbered segmental stripes and *Nv runt* for even-numbered segmental stripes, as summarized in Figure 8.”

It is not clear how the data presented justify this statement. The authors could combine morpholino treatments with *eve* stainings to test that hypothesis.

5) Section entitled ‘Expression and knockdown of *Nasonia eve*, *hairy* and *odd*’, last paragraph: It is difficult to understand what these sentences mean. How is a clock-like mechanism compatible with long-germ embryogenesis?

6) Section entitled ‘An ancestral role for eve in specifying posterior identity may be linked to growth zone behavior in short germ insects’: “In *Schistocerca*, *eve* is expressed in a posterior mesodermal domain and pair-rule stripes arise from this region, indicating that eve plays a role in posteriorness and not segmentation in basal insects.”

*Eve* is expressed in segmental patterns in spiders, myriapods and crustaceans, suggesting that the data from *Schistocerca* are not representative of the ancestral state.

7) 'Segmental' and 'pair-rule' stripes could be renamed stripes with single- and double-segment periodicity, for clarity, especially since pair-rule genes are sometimes expressed in stripes that have single-segment periodicity.

8) The authors point out the lack of 'segmental' stripes of odd (meaning stripes with single-segment periodicity). Have they checked whether one of the other paralogues of odd is expressed in the 'missing' stripes?

9) It would be useful if the authors could indicate the frequency of each phenotypic class in Figures 2 and 4.

10) It would be useful to know the identity of sequences (conventional gene names and, ideally, accession numbers) in the Figures showing sequence alignments and trees. For example, do the hairy-related genes included in the analysis include E(spl) orthologues?

11) Are the phylogenetic trees based on the entire aligned sequence, or on specific sequence blocks (domains) that could be aligned reliably?

12) Given the interest in comparing the arthropod and vertebrate segmentation clocks, the authors could comment on the absence of *hairy* oscillations in the Discussion. Have they looked at the expression of E(spl) orthologues?

13) First paragraph of Discussion: The authors reiterate the view that *Nasonia* develops using a long-germ mode, when in fact their paper argues that this is not the case (next paragraph). It would be interesting if the authors could step back and discuss in more general terms what their data suggests in terms of short- and long-germ mechanisms (see reviews by Davis and Patel, Ann Rev Entomol 2002, and Peel and Akam, Curr Biol 2003).

14) Section entitled ‘An ancestral role for eve in specifying posterior identity may be linked to growth zone behavior in short germ insects’: The concept of “posteriorness” is vague and should be explained.

15) The formation of posterior segments seems to be accompanied by a mild elongation the posterior part of embryo. This is another hallmark of short-germ development, which the authors could discuss.

16) Wilson and Dearden described the expression patterns and phenotypes for some of the same pair-rule genes in another hymenopteran, the honeybee. In the Discussion, it would be useful if the authors briefly compared the results from the honeybee with those from *Nasonia*.

---

## [Author Response]

*1) The paper contains a number of imprecise statements that should be corrected by careful editing of the manuscript. On several occasions the authors refer to cross-regulatory interactions among the pair-rule genes, based entirely on expression patterns (e.g., several points in the Discussion section). I think that is a weak basis for proposing such interactions. I recommend testing these hypotheses by combining morpholino treatments with the relevant in situs. Alternatively, the authors should make it clear that the proposed interactions are hypothetical. The remaining suggestions are highlighted in the minor comments below*.

We agree that the best experiments would be morpholino knockdown of each gene followed by in situ for each of the others, to really delineate the cross interactions that we propose. However, these experiments faced insurmountable technical limitations, and it was not possible to perform these studies. We have now made it more clear in the text that the proposed interactions represent a model that is hypothetical and based only on the suggestive expression patterns that we describe.

*2) The reviewers recommend separating, where possible, the description of data from the interpretations and the discussion that derives from it. Currently, much of the discussion is scattered within the Results section; in the Discussion the authors reiterate many of the same points*.

We have removed discussion from the Results section where possible (although in several places, we found it more appropriate to comment briefly on a result that will not be part of a general discussion), and have tried to ensure that there is no redundancy between the Results and Discussion. The Discussion is now shorter and smoother.

*Minor comments*:

*1) Introduction, 2nd paragraph: I am not aware of evidence suggesting that *odd* controls the oscillations, rather than being a read-out*?

We agree that the wording of our description of *odd-skipped* oscillations in *Tribolium* was not clear. This has been corrected and we now indicate that the expression of *odd* is indeed oscillating, but that the control of these oscillations is still not understood.

*2) Introduction, 2nd paragraph: Whether the common ancestors of bilaterians was segmented is still controversial*.

We are now more cautious and have added a reference to an excellent review discussing the competing models about the origin of segmentation, since it is difficult to concisely present these models in this context of the manuscript.

*3) When describing the cuticular phenotypes induced by morpholinos (*Figure 4
*and Results section), it is not always clear how the authors identify which of the segments are missing or affected? Are there distinctive cuticular features that identify each segment? These should be described*.

There are significant landmarks on the *Nasonia* cuticle that give clear indications of anterior segment identity. Spiracles are easy to detect on T2, A1, A2 and A3 and their position and order on critical segments allow most of the time their identification. Our interpretation of the phenotypes is based on the phenotypic series that we obtain through morpholino knockdown; we can often observe the reduction of structures or segments stepwise in this fashion. However, there are sometimes ambiguities. We now describe these landmarks in the text.

*4) Section entitled ‘*Nasonia runt* expression’: “Altogether, our data support a model in which *Nv eve* segmental stripes are established through the complementary action of *Nv odd* for odd numbered segmental stripes and *Nv runt* for even-numbered segmental stripes, as summarized in*
Figure 8*.*”

*It is not clear how the data presented justify this statement. The authors could combine morpholino treatments with eve stainings to test that hypothesis*.

We are unable to carry out this experiment with the very small number of injected embryos that are both non-uniformly developmentally delayed by the injection and also cannot be reliably recovered for subsequent staining without losing their structure. We have added a qualifying statement to indicate that this model is still theoretical.

*5) Section entitled ‘Expression and knockdown of* Nasonia eve, hairy *and* odd*’, last paragraph: It is difficult to understand what these sentences mean. How is a clock-like mechanism compatible with long-germ embryogenesis*?

It is difficult to imagine a clock-like mechanism as co-existing with an extreme long-germ insect, like *Drosophila*, where there is very little (or no) material that remains to be patterned/segmented after gastrulation. However, only *Drosophila* fits with a very restrictive definition of long germ.

Insect embryogenesis comprises a spectrum of short-germ to long-germ, and, although insects like *Nasonia* and *Apis* can be called long germ, they are less completely so than *Drosophila*, indicating that these two mechanisms do appear to co-exist, which is the major conclusion of our paper. It seems reasonable to imagine that evolution sampled more derived, long-germ embryogenesis types (with more of the germ segmented before cellularization) multiple times, while still relying on the ancestral mechanisms of segment patterning to deal with the remaining late-forming segments, even as the number of remaining segments varied. These two mechanisms would have had to co-exist at some point in the transition from the most extreme short germ embryo to the most extreme long-germ embryo. Our interpretation is that embryos like *Nasonia* represent a snapshot of that transition.

*6) Section entitled ‘An ancestral role for eve in specifying posterior identity may be linked to growth zone behavior in short germ insects’: “In* Schistocerca*, eve is expressed in a posterior mesodermal domain and pair-rule stripes arise from this region, indicating that eve plays a role in posteriorness and not segmentation in basal insects.*”

Eve *is expressed in segmental patterns in spiders, myriapods and crustaceans, suggesting that the data from* Schistocerca *are not representative of the ancestral state.*

A discussion of this point has been added to the text. While it is certainly true that several other arthropods express *eve* in segmental stripes, it is also true that other organisms, including vertebrates, have a posterior domain of *eve* expression like *Nasonia*. We discuss both of these points now in the text and leave open any conclusion about which is the ancestral state.

*7) 'Segmental' and 'pair-rule' stripes could be renamed stripes with single- and double-segment periodicity, for clarity, especially since pair-rule genes are sometimes expressed in stripes that have single-segment periodicity*.

We had struggled with this and clearly the reviewers also noted this. We have changed the name of the stripes to double-segment and single-segment throughout the descriptions of each gene.

*8) The authors point out the lack of 'segmental' stripes of odd (meaning stripes with single-segment periodicity). Have they checked whether one of the other paralogues of odd is expressed in the 'missing' stripes*?

We attempted to amplify and clone each of the three *odd* paralogs from *Nasonia* cDNA to look at this, using cDNA from staged animals (embryo, larva, adult). We were able to clone only the one described in the paper from embryo cDNA and one other (“*oddbowl A”* in the tree) from adult cDNA. The third was either expressed at levels too low to efficiently amplified, or may not be expressed. The expression of *oddbowl* A was not tested systematically in embryos, though a small-scale experiment revealed no embryonic expression.

*9) It would be useful if the authors could indicate the frequency of each phenotypic class in*
Figures 2 and 4.

Figure 2 does not show phenotypic classes but representative stainings of embryos at different stages of development. The pRNAi experiments with gap genes have been extensively described in several previous publications (Pultz et al., 1999, 2005; Olesnicky et al., 2006, Brent et al., 2007).

We now describe the frequency of the different phenotypic classes presented in Figure 4 obtained with morpholinos, which are approximately as follows:Eve:B 20.4%C 24%D 30.1%E 25%Odd:G 27.7%H 22.3%I 29.2%J 20.8%Hairy:L 13.4%M 25.0%N 36.6%O 25.0%

These values also now appear in the Materials and methods section of the paper.

*10) It would be useful to know the identity of sequences (conventional gene names and, ideally, accession numbers) in the Figures showing sequence alignments and trees. For example, do the hairy-related genes included in the analysis include E(spl) orthologues*?

The hairy-related genes include one possible E(spl) ortholog. The accession numbers for the sequences used in the alignments and trees are now given in the Materials and methods. Sequences in the *hairy* group have been updated, including longer sequences that have been released, in the database since the original submission, and their accession numbers are provided. The resulting alignment and tree now replace the original figures – the assignment of *Nvit H1* as the true *Nasonia hairy* is unaffected.

*11) Are the phylogenetic trees based on the entire aligned sequence, or on specific sequence blocks (domains) that could be aligned reliably*?

The phylogenetic trees were based on the entire aligned sequence; this is now mentioned in the figure legend.

*12) Given the interest in comparing the arthropod and vertebrate segmentation clocks, the authors could comment on the absence of *hairy* oscillations in the Discussion. Have they looked at the expression of E(spl) orthologues*?

We have not looked at the expression of E(spl) orthologues in *Nasonia.* We have added a brief comparison of arthropod and vertebrate clocks in the Discussion.

*13) First paragraph of Discussion: The authors reiterate the view that Nasonia develops using a long-germ mode, when in fact their paper argues that this is not the case (next paragraph). It would be interesting if the authors could step back and discuss in more general terms what their data suggests in terms of short- and long-germ mechanisms (see reviews by Davis and Patel, Ann Rev Entomol 2002, and Peel and Akam, Curr Biol 2003)*.

As discussed above, insects like *Nasonia* and *Apis* whose embryogenesis comprises a spectrum of short-germ to long-germ can be called long germ, though they are less completely so than *Drosophila (*which appears to be the only species that fulfill the strict definition of long germ). We agree with the reviewers that our work demonstrates that, unlike what we previously believed, *Nasonia* is not a strict long germ, indicating that these two mechanisms do appear to co-exist, which is the major conclusion of our paper. It seems reasonable to imagine that evolution sampled more derived, long-germ embryogenesis types (with more of the germ segmented before cellularization) multiple times, while still relying on the ancestral mechanisms of segment patterning to deal with the remaining late-forming segments, even as the number of remaining segments varied. These two mechanisms would have had to co-exist at some point in the transition from the most extreme short germ embryo to the most extreme long-germ embryo. Our interpretation is that embryos like *Nasonia* represent a snapshot of that transition.

*14) Section entitled ‘An ancestral role for eve in specifying posterior identity may be linked to growth zone behavior in short germ insects’: The concept of “posteriorness” is vague and should be explained*.

This concept, though vague, has been raised previously in the literature (see e.g., Ahringer J, *Genes Dev* 1996; Brown SJ et al., *Mech Dev* 1997). The significance of the posterior expression domains of *eve* in *Nasonia* and other arthropods is still not entirely clear, but it remains a possibility that it is in some way distinct from its expression in stripes and function in segmentation. The fact that its expression delineates a posterior growth zone in animals whose posterior segmentation occurs post-embryonically (e.g., Platynereis-de Rosa et al., *Evolution and Development* 2005) suggests that its ability to delineate a region of growth or patterning can be achieved independently of its striped expression and function (and that it may be an ancient feature). We wanted only to highlight that feature and with it, the possibility that it provides some additional, more general identity/information in the embryo posterior, though this explanation and characterization are still vague.

*15) The formation of posterior segments seems to be accompanied by a mild elongation the posterior part of embryo. This is another hallmark of short-germ development, which the authors could discuss*.

We didn’t emphasize the mild elongation of the posterior part of the embryo, since compared to the significant posterior growth in more short-germ insects, this elongation is modest in *Nasonia*. It is also difficult to distinguish what may be modest elongation in a short-germ sense vs simply the elongated appearance of a germband elongated (and retracted) embryo. We don’t feel that the data are sufficient to support further discussion.

*16) Wilson and Dearden described the expression patterns and phenotypes for some of the same pair-rule genes in another hymenopteran, the honeybee. In the Discussion, it would be useful if the authors briefly compared the results from the honeybee with those from* Nasonia.

An additional comparison of *Nasonia* pair rule genes to those of *Apis* has been added in the second paragraph of the Discussion.